# MC-SJD : Maximal Coupling Speculative Jacobi Decoding for Autoregressive Visual Generation Acceleration

## Abstract

While autoregressive (AR) modeling has recently emerged as a new paradigm in visual generation, its practical adoption is severely constrained by the slow inference speed of per-token generation, which often requires thousands of steps to produce a single sample. To address this challenge, we propose MC-SJD, a training-free, lossless parallel decoding framework designed to accelerate AR visual generation by extending the recently introduced Speculative Jacobi Decoding (SJD). Although SJD shows strong potential for accelerating AR generation, we demonstrate that token instability across iterations significantly reduces the acceptance rate, a limitation that primarily arises from the independent sampling process used during draft token generation. To overcome this, we introduce MC-SJD, an information-theoretic approach based on coupling, which substantially accelerates standard SJD by maximizing the probability of sampling identical draft tokens across consecutive iterations, all while preserving its lossless property. Remarkably, this method requires only a single-line modification to the existing algorithm, yet achieves substantial performance gains, delivering up to a $\sim 3.8\times$ speedup in image generation and $\sim 10\times$ speedup in video generation compared to standard AR decoding, without any degradation in output quality.

## 1 Introduction

Recently, autoregressive (AR) modeling has emerged as a cornerstone of modern generative AI Brown et al. (2020); Achiam et al. (2023), achieving state-of-the-art performance not only in text generation Touvron et al. (2023) but also across diverse modalities including images Liu et al. (2024); Sun et al. (2024a), video Agarwal et al. (2025), 3D meshes Weng et al. (2025), audio Du et al. (2024); Wang et al. (2023), and even robotics Pertsch et al. (2025). Its key strength lies in the ability to unify training and inference across modalities within a single framework, enabling flexible generation, editing, and translation. This cross-domain unification allows models to leverage rich knowledge from different sources, enhancing both understanding and generation Zhang et al. (2025).

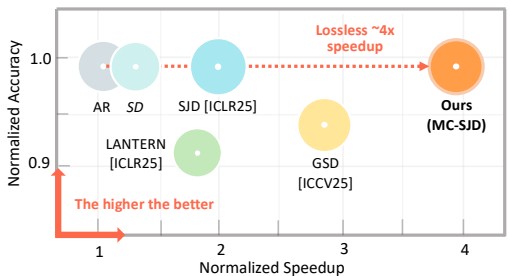

Figure 1: Comparison of recent SD methods for AR image generation. While recent works suffer from limited acceleration or sacrifice the quality, our **MC-SJD** achieves up to $\sim 4\times$ speedup over standard AR without any quality degradation.

However, the practical power of AR modeling is often constrained by the inherent cost of massive computation and exacerbated memory bottlenecks. Generating a sequence of $N$ tokens requires $N$ AR forward passes, leading to significant latency. The problem becomes particularly severe for high-dimensional data such as images and video, where thousands of tokens are needed to represent a single high-resolution instance Van Den Oord et al. (2017). This limitation acts as a critical barrier to the real-world deployment of multimodal AR models at scale.

Recently, speculative decoding (SD) (Leviathan et al. (2023)) has been actively explored (Sun et al. (2023); Yin et al. (2024)) to solve this problem, particularly for large language models (LLMs) in text generation. The core idea is to use a smaller, computationally inexpensive draft model to propose multiple candidate tokens, which are then verified in parallel by the more powerful target model. More importantly, SD is a ***lossless*** acceleration method, guaranteeing that its output distribution remains theoretically identical to standard AR sampling (Leviathan et al. (2023)). However, despite its effectiveness, SD has notable drawbacks: the overhead of training a separate draft model (Cai et al. (2024)), and limited performance in vision generation tasks (So et al. (2025); Jang et al. (2024)).

To address these problems, the pioneering Speculative Jacobi Decoding (SJD) Teng et al. (2024) was proposed, combining Jacobi iteration Song et al. (2021) with the stochastic verification criterion of SD. Briefly, SJD uses the output distribution from its own previous verification step as the draft for the next. This "Self-SD" approach eliminates the need for a separate, trained draft model, thereby resolving the idle-time bottleneck and demonstrating significant speedups, especially in image generation. However, while SJD shows promise, it delivers only about $\sim 2\times$ speedup in image generation, relatively modest compared to state-of-the-art SD methods in text, which achieve over $4\times$ acceleration Cai et al. (2024).

In this paper, we demonstrate that this problem can be solved with a simple tweak to the SJD process, offering an incredibly high speedup while maintaining the lossless property of SD. Our key finding is that the performance of the SJD is significantly limited by instability in its draft token sampling, leaving substantial room for further improvement. To unlock the potential of SJD, we introduce a simple yet highly effective, information-theory-inspired idea: **Coupling** Lindvall (2002). Specifically, we propose to couple the draft sampling process between consecutive Jacobi iterations, thereby increasing the probability of re-sampling the same tokens to promote stability. This method requires only a single-line modification to the standard SJD, making it extremely simple to implement without any additional training. Despite its simplicity, we demonstrate that our method significantly enhances the acceptance rate of SJD, enabling a remarkable lossless speedup of $\sim 3.8\times$ for AR image generation and an $\sim 10\times$ for video generation, compared to standard AR decoding.

## 2 PRELIMINARIES

**Notation.** We denote by $X_i^t$ the token at the $i$-th position of a sequence $X$ at Jacobi iteration $t$ (defined later). When clear from context, we omit the subscript/superscript $i$ or $t$ to refer to the entire sequence or to the collection of distributions, respectively. Similarly, we denote by $p_i^t(\cdot)$ the token distribution at position $i$ in iteration $t$. We assume all distributions are on the same support $V$.

### 2.1 SPECULATIVE DECODING

The main goal of Speculative Decoding (SD) (Leviathan et al. (2023)) is to reduce the number of sequential calls of target model $p$ while ensuring that final outcome matches the AR sampling distribution, $\prod_i p_i(x \mid X_{<i})$. Specifically, we assume two models: a target $p(x)$, which we wish to accelerate, and a draft $q(x)$, which is faster than $p(x)$ but less accurate. SD proceeds as follows:

1. **Drafting**: Sample $L$ draft tokens from the draft distributions, $X_{i:i+L-1} \sim q_{i:i+L-1}(\cdot)$.

2. **Evaluate**: In parallel, have the target model evaluate the token probabilities along the drafted prefixes, i.e., $\left\{ p_j(X_j \mid X_{<j}) \right\}_{j=i}^{i+L-1}$.

3. **Verify**: Run Algorithm 3 with $(p_i, q_i, X_i)$ sequentially until a rejection occurs (i.e., the procedure returns $k = 0$); accept all previously verified tokens.

4. **Repeat**: If the generation is not yet complete, return to **Drafting** and repeat the process.

Transformers natively support the parallel evaluation in step (2) via masked attention, ideally in $O(1)$ sequential depth. Thus, if *acceptance* occurs in step (3), the procedure may emit multiple tokens in effectively $O(1)$ sequential time, reducing the total NFEs compared with standard AR decoding.

The Sampling of Algorithm 3 guarantees that even if the input is $X \sim q(\cdot)$, the output $Y$ returned by the algorithm satisfies $Y \sim p(\cdot)$ Chen et al. (2023). Because each Markov chain follows valid sampling from $p$ until the first rejection occurs, the theoretical correctness of speculative decoding

**Algorithm 1** Speculative Jacobi Decoding

**Require:** AR Model $p_\theta$, draft Length $L$, Max Sequence $N$
1: $p_t^i \leftarrow$ `Random()`
2: $X_i^t \sim p_i^t \quad \forall i, t$ $\qquad \triangleright$ Initialize
3: **while** $i < N$ **do**
4:     **parallel for** $j = i$ to $i + L$ : $\triangleright$ Drafting
5:        $\textcolor{red}{X_j^t \sim p_j^t(x)}$
6:     **parallel for** $j = i$ to $i + L$ : $\triangleright$ Evaluate
7:        $p_j^{t+1} \leftarrow p_\theta(\cdot \mid X_{0:j-1}^t)$
8:     **for** $j = i$ to $i + L$ : $\qquad \triangleright$ Verify
9:        $k, X_j^{t+1} \leftarrow$ MRS$(p_j^{t+1}, p_j^t, X_j^t)$
10:     **if** $k = 0$ : **break;**
11:    $i \leftarrow j, t \leftarrow t + 1$
12: **end while**
13: **return** $X$

**Algorithm 2** Pseudo Code for our **MC-SJD**

**Require:** AR Model $p_\theta$, draft Length $L$, Max Sequence $N$
1: $p_t^i \leftarrow$ `Random()`
2: $X_i^t \sim p_i^t \quad \forall i, t$ $\qquad \triangleright$ Initialize
3: **while** $i < N$ **do**
4:     **parallel for** $j = i$ to $i + L$ : $\triangleright$ Drafting
5:        $\dashrightarrow, X_j^t \leftarrow$ MRS$(p_j^t, p_j^{t-1}, X_j^{t-1})$
6:     **parallel for** $j = i$ to $i + L$ : $\triangleright$ Evaluate
7:        $p_j^{t+1} \leftarrow p_\theta(\cdot \mid X_{0:j-1}^t)$
8:     **for** $j = i$ to $i + L$ : $\qquad \triangleright$ Verify
9:        $k, X_j^{t+1} \leftarrow$ MRS$(p_j^{t+1}, p_j^t, X_j^t)$
10:     **if** $k = 0$ : **break;**
11:    $i \leftarrow j, t \leftarrow t + 1$
12: **end while**
13: **return** $X$

is ensured. As shown in Alg 3, the acceptance probability per token, $\min\{1, p(x)/q(x)\}$, is the key factor that determines the overall speedup. We formalize this in the following proposition:

**Proposition 1** *Let $q$ be the draft distribution and $x \sim q(x)$, then, final output from* MRS*(Alg.3) strictly follow the distribution of target model $p(x)$. Moreover, the acceptance rate of this algorithm is defined as*

$$\mathbb{E}_{x \sim q(x_i)} min(1, \frac{p(x)}{q(x)}) = 1 - \mathcal{D}_{TV}(p, q)$$

*where $\mathcal{D}_{TV}$ denotes total variation $\frac{1}{2} \sum_v |p(v) - q(v)|$.*

*Proof:* See appendix. Typically, standard SD methods employ a "cheaper" AR model to produce draft tokens/distributions. While this strategy has shown promising results in the text-generation domain Cai et al. (2024), it has several drawbacks: the need to train a separate draft model, communication bottlenecks between the draft and target models, and limited speedups in non-text AR generation domains So et al. (2025); Jang et al. (2024). These issues have hindered the adoption of SD techniques beyond text, limiting the potential of AR modeling across different modalities.

**Algorithm 3** MRS(p,q,x); Modified Rejection Sampling

**Input**: Distribution $P, Q$. Tokens $X \sim Q$
**Output**: Accept signal $k$, Random variable $Y$
1: Sample $u \sim \mathcal{U}[0, 1]$
2: **if** $u \leq \min(1, \frac{P(X)}{Q(X)})$
3:    **return** 1, Y=X
4: **return** 0, $Y \sim norm(max(0, P(x) - Q(x)))$

## 2.2 SPECULATIVE JACOBI DECODING

Speculative Jacobi Decoding (SJD) Teng et al. (2024) is a pioneering, training-free algorithm to solve the aforementioned problems of standard speculative decoding. As depicted in Algorithm 1, SJD eliminates the need for a separate draft model $q(\cdot)$. Instead, it leverages the probability distribution from its own previous validation step as the draft for the next iteration. This "Self-SD" approach does not impact the theoretical accuracy guarantees of speculative decoding, because the verification mechanism (Alg. 3) ensures the output is always a valid sample from the target model's distribution, regardless of the input draft. This framework makes the process highly efficient as it removes the overhead of training a separate model and eliminates the idle time where the target

model would wait for draft tokens. Due to these properties, SJD first achieves a $\sim$2x speedup in AR image generation domain, while retaining its lossless and training-free nature.

**Connection to the Fixed-Point Methods** Another key difference between SJD and other SD frameworks is that it *reuses* information from rejected tokens in the next drafting. This allows SJD to be framed as a fixed-point method Coddington et al. (1956) that updates the entire sequence $X^t$ at once via the iteration $X^{t+1} \leftarrow F(X^t)$, which is known to converge to a solution very fast Hutzenthaler et al. (2021), under the assumption of continuity of $X$ and contraction on $F$. SJD can be seen as practical variant of this, which relaxes the contraction property and is adapted for a discrete token space, by using a probabilistic convergence criterion from SD instead of a numerical difference $||X^t - X^{t-1}||$, which is ill-suited for the discrete case. In practice, while we want our sequence $X^t$ to be *refined* across iterations within the fixed-point framework, converging toward a solution, we have found that the current SJD has very little effect in this regard.

## 3 MOTIVATION AND ANALYSIS

Despite achieving a $\sim$2x speedup in image AR, we find that performance potential of SJD is *significantly limited* by the variance introduced during its stochastic draft sampling process. To gain an intuitive understanding of this, we start with an analysis of the acceptance rate of SJD. At iteration $t$ in SJD, the target distribution is $p^t(\cdot)$ and the draft distribution is $p^{t-1}(\cdot)$. As noted in Proposition 1, the acceptance rate can be expressed in terms of the Total Variation, as follows :

$$\beta_i^{(t)} = 1 - \mathcal{D}_{TV}\left(p_i^{(t)}(x), p_i^{(t-1)}(x)\right) \tag{1}$$

$$= 1 - \mathcal{D}_{TV}\left(p_\theta\left(\cdot \,\Big|\, X_{<i}^{(t-1)}\right), p_\theta\left(\cdot \,\Big|\, X_{<i}^{(t-2)}\right)\right), \tag{2}$$

where $p_\theta$ denotes the autoregressive model and $X_{<i}$ denotes the prefixes $\{X_{i-1}, X_{i-2}, \dots\}$. As shown in Eq. 2, The acceptance rate $\beta_i^{(t)}$ is directly influenced by the *context change* between iterations $t-1$ and $t-2$. In other words, the acceptance rate for token $i$ is driven by changes in its *prefixes*, including both previously accepted tokens but also the other rejected tokens in the draft. This leads directly to the following observation:

**Observation 1** *High context similarity between consecutive drafts tends to yield a higher speedup.*

This can be easily validated by Eq. 2: the greater the similarity between the contexts $X_{<i}^{(t-1)}$ and $X_{<i}^{(t-2)}$, the more similar their corresponding output distributions $p_\theta(\cdot|X_{<i})$ will be (under a mild Lipschitzness assumption on the model's logits from tokens). This results in a lower TV distance and, consequently, a higher acceptance rate $\beta$.

We also empirically validate it in Fig. 2, plotting the 300 independent samples with their mean number of changed tokens between consecutive sequence drafts (Hamming distance) against the total number of function evaluations (NFE) required for SJD generation. As shown, there is a strong correlation between these two metrics, indicating that context similarity plays a crucial role for faster generation in SJD.

However, despite this correlation, Fig. 2 shows that the average number of token difference is approximately 94% tokens - 60 of window size 64 -, indicating significantly large portion of tokens are changed in each iteration. We observe that this high degree of change not only critically limits SJD's single iteration acceptance rate but also poses a more severe problem if we consider behavior on multiple consecutive iterations, the *convergence* of SJD.

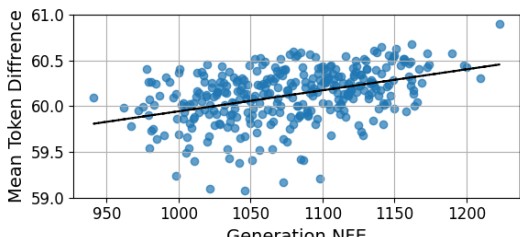

Figure 2: Generation NFE v.s Mean Token Difference during SJD with window size $L = 64$. As shown, a sample that is generated with smaller NFE tends to have small mean token difference.

**Observation 2** *The per-token acceptance rate $\beta_i^t$ during the SJD process exhibits high variance and does not show converging behavior.*

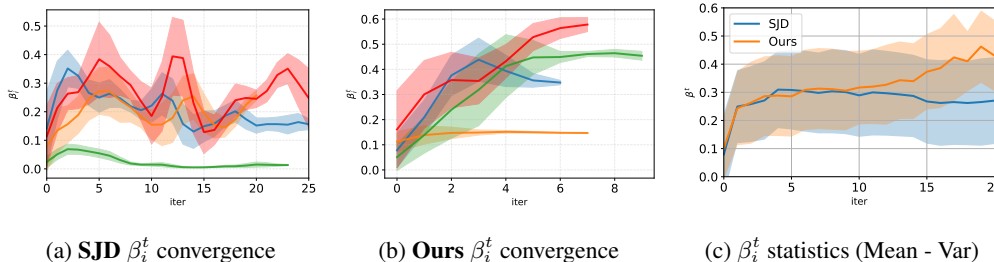

(a) **SJD** $\beta_i^t$ convergence      (b) **Ours** $\beta_i^t$ convergence      (c) $\beta_i^t$ statistics (Mean - Var)

Figure 3: (a), (b) The trajectory of tokenwise acceptance rate $\beta_i^t$ during the jacobi iterations (a) Standard SJD shows most tokens have large variation during iteration and do not exhibit improvement behavior. (b) After applying our coupled sampler $\pi_{MC}$. Now most of tokens has very small fluctuation, showing general upward trends. (c) Mean and variance of $\beta_i^t$ across all token index. While standard SJD does not show improvement, ours shows clear upward, refining behavior.

Ideally, the acceptance rate for a given token should increase over the SJD iterations. As the left-most context becomes filled with stable, accepted tokens, an "improvement signal" should propagate to the right, progressively enhancing the quality of the draft sequences. However, our empirical results reveal the opposite behavior. Fig. 3(a) plots the trajectory of $\beta_i^t$ for representative tokens, showing that the acceptance rate frequently fluctuates without any consistent upward trend. This instability is further confirmed in Fig. 3(c) (blue line), which aggregates the statistics across all tokens. After an initial jump, the mean acceptance rate not only remains low but also fails to improve, exhibiting random fluctuations with high variance throughout the process.

### 3.1 ANALYSIS

We then investigate the root cause of this low context similarity. Since the context sequences $X^{(t)}$ are realizations of random variables drawn from $p^{(t)}(\cdot)$ at each iteration, a natural way to quantify their similarity is by measuring the **collision probability**, defined as $\Pr[X_i^{(t)} = X_i^{(t-1)}]$. As described in Algorithm 1, the drafting stage of SJD (Line 5) samples the draft token $X_i^{(t)}$ *independently* from its distribution $p_i^t(x)$. In this independent sampling scheme, the collision probability between $X_i^{(t)}$ and $X_i^{(t-1)}$ can be analytically computed as follows:

**Proposition 2 (SJD Collision Probability)** *Standard SJD has the following collision probability for token $i$ at iteration $t$:*

$$C_{SJD}(p^{(t)}, p^{(t-1)}) = \sum_{x \in \mathcal{V}} p_i^{(t)}(x) \cdot p_i^{(t-1)}(x)$$

*where $\mathcal{V}$ denotes the vocabulary. This value is bounded as follows:*

$$C_{SJD}(p, q) \le e^{-1/2 \cdot (H_2(p) + H_2(q))}$$

*where $H_2(p) = -log(\sum_x p(x)^2)$ is the Rényi-2 entropy of $p$.*

*Proof:* See appendix. As shown, even when two distributions are similar, their collision probability is constrained by the (Rényi-2) entropy of the underlying distributions and vocabulary sizes. Unfortunately, unlike text AR models, visual AR models are known to generate very flat distributions So et al. (2025). This is because of the inherent redundancy in visual tokens and the complexity of visual patterns makes a large number of different tokens plausible continuations of a sequence.

We also visualize the empirical collision probability, $C_{SJD}$, during the SJD process in Fig. 4. Fig. 4 (a) shows most of its values remain at an extremely low value. and (c) illustrates that this value is nearly zero regardless of whether the TV distance is small. Consequently, the standard SJD propagates different contextual information to subsequent tokens in each iteration, significantly destabilizing the iteration and causing the convergence to fluctuate unpredictably. We identify this discrepancy between the proximity in probability space and the realized token space as the *key factor* limiting the speedup in SJD.

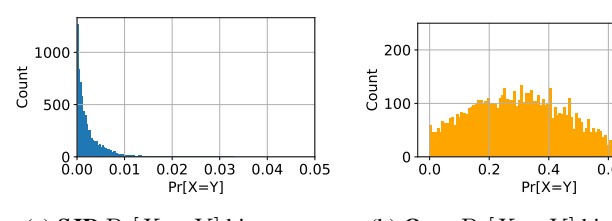 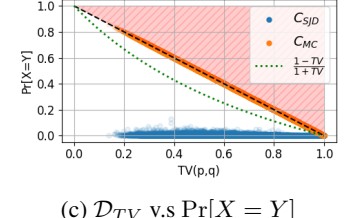

(a) **SJD** $\Pr[X = Y]$ histogram     (b) **Ours** $\Pr[X = Y]$ histogram     (c) $\mathcal{D}_{TV}$ v.s $\Pr[X = Y]$

Figure 4: Visualization of Collision probabilities. (a) During standard SJD, $C_{SJD}$ are concentrated on extremely small values. (b) Our Coupler elevates this to much higher values, significantly enhancing the context similarity. (c) Standard SJD has a low $\Pr[X = Y]$ even when the corresponding TV distance is low. The green dot-line denotes the $\pi_{GS}$ lower bound $\pi_{GS} \geq (1 - \mathcal{D}_{TV})/(1 + \mathcal{D}_{TV})$.

## 4 METHODS

Our main idea is that making ***Coupling*** (Lindvall (2002)) between the draft distributions from consecutive iterations can increase the collision probability without compromising the theoretical lossless correctness of the SJD. To formalize, we begin with mathematical definition of coupling :

**Definition 1 (Coupling)** *For two distributions $P(\cdot)$ and $Q(\cdot)$ on the same support $\mathcal{V}$, a joint distribution $\pi(\cdot, \cdot)$ over $\mathcal{V} \times \mathcal{V}$ is a **Coupling** of $P$ and $Q$ if its marginals satisfy:*

$$\sum_{y \in \mathcal{V}} \pi(x, y) = P(x) \quad and \quad \sum_{x \in \mathcal{V}} \pi(x, y) = Q(y)$$

The key insight lies in the marginalization property of a coupling. If we sample a pair of variables from a joint distribution $\pi(x, y)$, the marginal distribution of each individual variable remains identical to its original distribution (e.g., $P(x)$ and $Q(y)$). Therefore, using a token sampled from a coupling is a provably valid replacement for independent sampling within the SJD framework. We formally stated this in the following theorem:

**Theorem 1** *Let $\Pi_i^{(t)}$ be the set of all possible couplings between $p_i^{(t)}$ and $p_i^{(t-1)}$. If we sample a pair $(X_i^{(t)}, X_i^{(t-1)}) \sim \pi(\cdot, \cdot)$ for any $\pi \in \Pi_i^{(t)}$ and use $X_i^{(t)}$ as the draft token in Algorithm 1, the final output distribution still correctly matches the target model's distribution.*

*Proof Sketch:* See appendix. For any given coupling $\pi$, we can define its effectiveness using a metric called the **Coupling Cost**, denoted $C(\pi)$. This cost measures the probability of sampling identical variables from the joint distribution, which is the same metric we previously referred to as the collision probability:

**Definition 2 (Coupling Cost)** *Let $\pi_{P,Q}$ be a coupling of distributions $P$ and $Q$ as per Definition 1. The Coupling Cost is defined as:*

$$C(\pi_{P,Q}) = \Pr_{(X,Y) \sim \pi_{P,Q}} [X = Y] = \mathbb{E}_{(X,Y) \sim \pi_{P,Q}} \mathbb{1}\{X = Y\}$$

From this perspective, the standard SJD process can be understood as using an *independence coupling*, where $\pi_{SJD}(x, y) = p_i^{(t)}(x) \cdot p_i^{(t-1)}(y)$ and the cost of this particular coupling is $C(\pi_{SJD}) = \sum_v p_i^{(t)}(v) p_i^{(t-1)}(v)$, a value we have already shown to be extremely low in AR image generation. Finally, our main objective can be safely reframed as finding an alternative coupling, $\pi^*$, that **maximizes** this cost, thereby promoting context similarity without compromising the exactness guarantee of the framework. We next present alternative couplings that achieve this objective.

### 4.1 MAXIMAL COUPLING

Consider the computation graph of $X^t$ and $p^t(x)$ during the SJD process :

$$X^{t-2} \rightarrow p^{t-1}(X) \rightarrow X^{t-1} \rightarrow p^t(X) \rightarrow X^t \tag{3}$$

As shown, at the time of sampling step for $X^{(t)}$, we have access to the full information of two probability distributions $p^{(t)}$ and $p^{(t-1)}$. As is well-established in many literature on information theory and optimal transport Villani et al. (2008); Bavarian et al. (2016), *having complete information of both distributions allows us for the construction of a **maximal coupling***, which has the cost of $c(\pi_{p,q}) = 1 - \mathcal{D}_{TV}(p, q)$ that any two distribution can maximally have.

In Algorithm 2, we present the implementation of SJD with draft sampling with maximal coupling process. As shown, the only modification required is in the drafting phase (Line 4), where we now sample the draft token $X^t$ using a *coupled sampler* instead of sampling it independently from $p_i^t(\cdot)$. Interestingly, as shown, the implementation of this coupling is exactly identical with the modified rejection sampling, $\text{MRS}(\cdot)$ (Alg.3) which we used for speculative decoding verification process. This can be easily validated by the fact that $\text{MRS}(\cdot)$ returns $Y \sim P$ from an input $X \sim Q$, ensuring that the marginals of the generated pair $(Y, X)$ match $P$ and $Q$, which satisfies the definition of a coupling. Moreover, as established in Proposition 1, the acceptance rate $\text{MRS}(\cdot)$ - probability of $\Pr[X = Y]$ - is $1 - \mathcal{D}_{TV}(p, q)$. This value is the theoretical upper bound of coupling cost, confirming that this procedure constitutes a maximal coupling. We formally state this as follows:

**Theorem 2** *Let the pair (X,Y) be generated by Algorithm 3. Then, their resulting joint distribution $(X, Y) \sim \pi_{MC}$, is a valid coupling of P and Q. Moreover, its coupling cost,*

$$C(\pi_{MC}) = 1 - \mathcal{D}_{TV}(P, Q)$$

*is the upper bound for the cost of any $\pi \in \Pi$ with P and Q.*

As illustrated in Fig. 4 (c), this upper bound, represented by the black dashed line, shows a significant gap compared to the coupling cost of standard SJD ($C_{SJD}$). Applying maximal coupling within SJD, elevates this low values to their upper bound (orange dots), thereby strongly promoting high context similarity and achieving a greater speedup. We also show the distribution of $C(\pi_{MC})$ in Fig. 4 (b). In Fig. 3 (b), (c), we show the trajectories and statistics of the $\beta^t$, during iterations with MC-SJD. As shown, most tokens now exhibit minimal fluctuation with a general upward trend, resulting in a much higher overall acceptance rate compared to standard SJD, leading to lower NFEs.

---

**Algorithm 4** $\text{GS}(P, Q, G)$; Gumbel Noise Sharing

---

**Input**: Distributions $P, Q$ over a vocabulary $\mathcal{V}$. A shared Gumbel noise vector $G = (g_1, \ldots, g_{|\mathcal{V}|})$ where $g_i \sim \text{Gumbel}(0, 1)$.
**Output**: A coupled pair of random variables $(X, Y)$.
  1:  $X \leftarrow \text{argmax}_{i \in \mathcal{V}}(\log(P_i) + g_i)$                          ▷ Sample from P using G
  2:  $Y \leftarrow \text{argmax}_{i \in \mathcal{V}}(\log(Q_i) + g_i)$                  ▷ Sample from Q using the same G
  3:  **return** $(X, Y)$

---

## 4.2 GUMBEL COUPLING

While maximal coupling is theoretically optimal in terms of coupling cost, we also introduce a simpler alternative, ***Gumbel Coupling***, which is easier to implement but achieves a comparable coupling cost with maximal coupling. We denote $\pi_{GS}$ in Algorithm 4 and SJD implementation with this coupling in Algorithm 5. As shown, this algorithm is based on the Gumbel-Max trick that relies on sharing the same random noise vector to couple two categorical sampling processes. We first establish its validity and provide a lower bound for its cost:

**Theorem 3** *Let the pair (X,Y) be generated by Algorithm 4. Then, their resulting joint distribution $(X, Y) \sim \pi_{GS}$, is a valid coupling of P and Q. Its worst-case coupling cost is lower-bounded by:*

$$C(\pi_{GS}) \geq (1 - \mathcal{D}_{TV}(P, Q))/(1 + \mathcal{D}_{TV}(P, Q))$$

*Proof sketch:* The coupling validity of $\pi_{GS}$ can be easily shown based on the Gumbel-Max Trick Gumbel (1954), where this trick known to yield output that follows input categorical distribution. In this setting, this lower bound has been well-studied in recent works Bavarian et al. (2016) □. As shown in Fig. 4 (c), this lower bound is significantly greater than the independence coupling $\pi_{SJD}$ and almost tight to the optimal $1 - \mathcal{D}_{TV}$, thus achieving performance comparable to $\pi_{MC}$. Moreover, $\pi_{GS}$ can even yield slightly better NFEs in some tasks. Since this lower bound is applicable to **any pair of distributions** during an iteration, this *Gumbel Coupling* promotes more long-range stabilization, whereas $\pi_{MC}$ optimizes greedily in each consecutive iteration, and this behavior can be slightly beneficial depending on the task. We will discuss in the next section.

Table 1: Evaluation results of AR Image generation model, Lumina-mGPT, on MS-COCO dataset.

| Configuration | | NFE (↓) | Latency (↓) | Acceleration (↑) | | FID (↓) | IS (↑) | CLIP-Score (↑) |
|---|---|---|---|---|---|---|---|---|
| | | | (A100) | NFE | Latency | | | |
| A | *Vanilla AR* | 2390 | 102.03s | 1.00× | 1.00x | 30.79 | 32.81 | 31.31 |
| B | SJD (L=16) | 1058.6 | 43.02s | 2.25× | 2.37x | 30.77 | 32.78 | 31.32 |
| D | + **Ours** ($\pi_{MC}$) | **814.5** | **32.75s** | **2.94×** | **3.12x** | 30.73 | 33.56 | 31.32 |
| D | + **Ours** ($\pi_{GS}$) | 819.4 | 32.89s | 2.92× | 3.10x | 30.78 | 32.77 | 31.37 |
| B | SJD (L=32) | 1031.2 | 42.99s | 2.32× | 2.37x | 30.78 | 32.82 | 31.31 |
| D | + **Ours** ($\pi_{MC}$) | 666.0 | 26.77s | 3.59× | 3.81x | 30.79 | 33.56 | 31.32 |
| D | + **Ours** ($\pi_{GS}$) | **652.3** | **26.44s** | **3.66×** | **3.86x** | 30.75 | 32.91 | 31.39 |
| B | SJD (L=64) | 1035.9 | 42.98s | 2.31× | 2.37x | 30.81 | 32.76 | 31.31 |
| D | + **Ours** ($\pi_{MC}$) | **567.7** | 24.41s | **4.21×** | 4.18x | 30.83 | 33.43 | 31.37 |
| D | + **Ours** ($\pi_{GS}$) | 568.0 | **24.24s** | 4.21x | **4.21x** | 30.90 | 32.80 | 31.37 |
| C | GSD (L=32,G=3) | 925.9 | 38.98s | 2.58× | 2.62x | 31.50 | 29.76 | 31.33 |
| C | GSD (L=32,G=10) | 701.4 | 29.13s | 3.40× | 3.50x | 33.21 | 26.78 | 31.25 |

Table 2: Video generation results on Cosmos1-AR-4B, on real-state-10k dataset.

| Metric | Vanilla | L=16 | | | L=32 | | | L=64 | | | L=128 | | |
|---|---|---|---|---|---|---|---|---|---|---|---|---|---|
| | AR | SJD | + $\pi_{MC}$ | + $\pi_{GS}$ | SJD | + $\pi_{MC}$ | + $\pi_{GS}$ | SJD | + $\pi_{MC}$ | + $\pi_{GS}$ | SJD | + $\pi_{MC}$ | + $\pi_{GS}$ |
| **NFE** (↓) | 7680 | 2272.8 | 1990.5 | **1940.6** | 1886.4 | 1293.7 | **1267.3** | 1802.3 | **835.9** | **810.7** | 1789.9 | 577.8 | **564.4** |
| **Latency (s)** (↓) | 157.25 | 54.12 | 48.93 | **47.97** | 48.43 | 32.36 | **32.01** | 48.19 | 22.38 | **21.58** | 47.73 | 15.87 | **13.60** |
| **FVD** (↓) | 156.9 | 157.1 | 159.3 | 154.8 | 153.2 | 155.8 | 153.6 | 163.6 | 155.8 | 152.9 | 158.3 | 157.8 | 152.4 |

## 5 EXPERIMENTAL RESULTS

In experiments section, we mainly focus on validating two aspects : (i) How much acceleration can we gain by applying our method atop SJD, (ii) Does our algorithm truly preserve generation quality, although we show it theoretically.

**Setup** Similar to original SJD paper, we mainly evaluate with Lumina-mGPT Liu et al. (2024) for AR image generation. We also evaluate our method with the more SOTA AR Image model, Janus-Pro Chen et al. (2025), to validate our method's generalization. Moreover, beyond image generation, we also evaluate with an AR video generation model, cosmos-ar Agarwal et al. (2025), which has longer generation sequence length and expected to have more redundancy. The more detailed settings are in the appendix.

**Metrics and Datasets** To evaluate the quality, we measured FID Heusel et al. (2017) , which denotes the distribution distance compared to reference and generated datasets, IS Barratt & Sharma (2018) and CLIP score Radford et al. (2021) for fair comparison. To evaluate speed, we measure number of function evaluation (NFE), which indicates the number of sequential forward steps, and the real latency on 1x NVIDIA A100 device. We mainly use MS-COCO (val) dataset for image generation evaluation and real-state-10k for video generation. More details are in appendix.

**Baselines:** We benchmarked our method against three baselines: (A) standard autoregressive decoding (*Vanilla AR*), (B) Speculative Jacobi Decoding (SJD) (C) Grouped Speculative Decoding (GSD) So et al. (2025), which is recently proposed lossy SD methods for image generation and (D) Ours. We implement (C) and (D) atop (B) for fair comparison.

| Config | | NFE (↓) | Latency(s)(↓) | FID (↓) | IS (↑) |
|---|---|---|---|---|---|
| A | *Vanilla AR* | 576 | 13.218 | 37.96 | 22.39 |
| B | SJD (L=16) | 319.93 | 10.213 | 37.96 | 22.25 |
| D | + **Ours**($\pi_{MC}$) | **190.21** | 6.345 | 37.46 | 22.19 |
| D | + **Ours**($\pi_{GS}$) | **189.99** | 6.336 | 37.13 | 22.53 |
| B | SJD (L=32) | 318.01 | 10.582 | 37.76 | 21.80 |
| D | + **Ours**($\pi_{MC}$) | **154.76** | 5.471 | 38.34 | 22.17 |
| D | + **Ours**($\pi_{GS}$) | **154.42** | 5.388 | 37.49 | 22.43 |

Table 3: Janus-Pro (7B) on MS-COCO 2017.

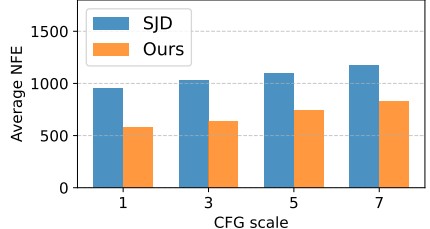

Figure 5: CFG scale vs. NFE. All experiments use Lumina-mGPT 768×768 (7B).

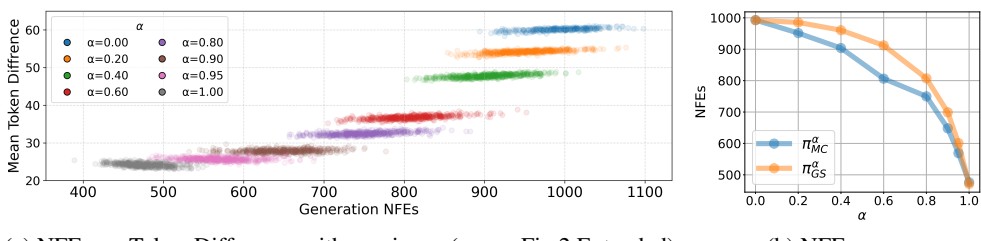

(a) NFEs v.s Token Difference with varying $\alpha$ ($\pi_{MC}$, Fig.2 Extended)  (b) NFEs v.s $\alpha$

Figure 6: Coupling Strength Experiments ($L = 32$). Increasing the coupling strength $\alpha$ decrases Mean Token Diffrence and Generatiation NFEs until $\alpha \approx 1$, indicating that pushing collision probability to its theoretical optimal tends to have significant positive effect on minimzing total NFEs.

**Results** Table 1 presents our main results for AR image generation on Lumina-mGPT. As shown, our method (D) accelerates the AR decoding (A) by up to ~4x and SJD (B) by ~1.8x without compromising its exactness guarantee, maintaining identical FID, IS and CLIP scores. Notably, while standard SJD fails to achieve a meaningful speedup with an increased window size (L), our MC-SJD demonstrates consistent acceleration as the window size grows, strongly suggesting that our coupling helps to stabilize SJD's convergence. Finally, compared to (C) , lossy SD method GSD, while it also significantly reduces the NFE, it results in a degradation of the FID and CLIP scores. Our method, in contrast, shows an even faster speedup than lossy GSD while maintaining quality exactness. In Table 3, we also report results on the SOTA AR image model, Janus-Pro (7B). As shown, our method consistently accelerates the standard SJD process by up to 2.1x, achieving a final step compression of 3.7x.

In Table 2, We depict the quantitative results on AR video generation model, cosmos-ar (4B). Remarkably, as shown, applying our method achieves ~10x actual acceleration with no loss in performance. This large acceleration gain mainly stems from the strong temporal redundancy between consecutive frames in video AR generation, and we believe our results will unlock huge potential in this field, where research progress has recently been hindered by speed bottlenecks.

### 5.1 FURTHER ANALYSIS ON BEHAVIOR OF COUPLED SAMPLING

**Coupling Strength** To more precisely understand what our coupling affects the convergence of SJD, we introduce a notion of *coupling strength* by interpolating between the independent distribution of SJD ($\pi_{SJD}$) and our coupling joint distribution ($\pi_{cpl}$). We define the $\alpha$-coupling by,

$$\pi_{\text{cpl}}^{\alpha}(x,y) = \alpha \cdot \pi_{\text{cpl}}(x,y) + (1-\alpha) \cdot \pi_{\text{ind}}(x,y), \tag{4}$$

Thus $\alpha = 0$ recovers vanilla SJD, and $\alpha = 1$ corresponds to ours. We present validity proof and implementation details of this coupling in in Appendix A.5.

In Fig.6, (a) we plot the mean token difference between consecutive iteration against the resulting NFE by changing $\alpha$, and (b) summarizes the average NFE on $\alpha$. We observe a clear monotonic trend: as $\alpha$ increases, both the mean Hamming distance and the NFE consistently decrease, and this improvement continues up to $\alpha \approx 1$. This confirms our hypothesis from Sec.3 that increasing the collision probability in token space directly enhances context stability and, consequently, reduces the total number of function evaluations required for generation.

**Multi-Step Behavior** While we mainly focused on analyzing and improving theoretically available 1-step behavior of SJD, it is worth to investigate how our coupler affects the longer-step behavior of draft tokens. In Fig. 8, we visualize the Hamming distance (mean token difference) between $N$-step iterations, $\text{Hamm}(t, t + N)$, when our couplers are applied. As shown, for the 1-step case, $\pi_{MC}$ consistently has a smaller distance than $\pi_{GS}$, aligning with our theory. However, for multi-step ($N = 2, 3$), this relationship is **reversed** in the high coupling-strength regime. This occurs because, while $\pi_{MC}$ is optimal for maximizing the 1-step collision probability, there is no non-trivial bound on its multi-step, whereas $\pi_{GS}$ have a lower bound of $1 - \mathcal{D}_{TV}/(1 + \mathcal{D}_{TV})$ between any pair of multi-step iterations, as shown in Theorem 3. We interpret this as $\pi_{GS}$ having better *long-range stability* than $\pi_{MC}$, and this can be advantageous in tasks where draft prediction is relatively easy, so that keeping early draft tokens unchanged can have more beneficial effect on the final convergence (smaller NFEs) than continually updating them with slightly more accurate information, as is the case in video generation(Tab. 2), which requires only slight modification from given start frame.

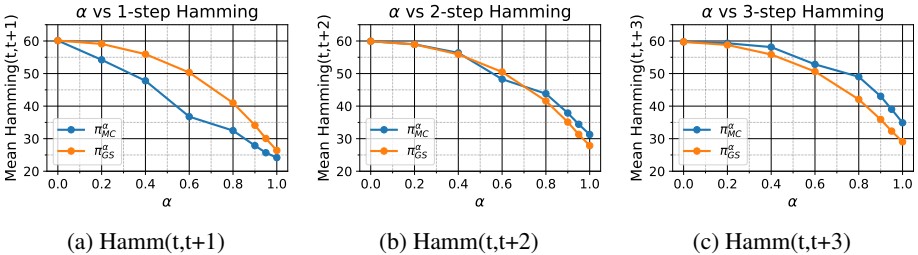

Figure 7: Qualitative comparison between **Ours** v.s. *AR* on Lumina-mGPT. (zoom-in to view).

(a) Hamm(t,t+1)  (b) Hamm(t,t+2)  (c) Hamm(t,t+3)

Figure 8: Multi step behavior of draft tokens when our $\pi_{MC}$ and $\pi_{GS}$ applied.

## 5.2 ABLATION STUDIES

**Coupling Overhead.** Our coupler can be implemented with almost no overhead compared to SJD. In efficient implementation, we can simply vectorize verification loop(Alg.2, Line 8) without *break* and save the index where the first rejection ($k = 0$) occurs, thereby integrating MC-sampling drafting (Alg. 2, Line 5) with verification into a single operation to minimize overhead. Similarly, the Gumbel noise in $\pi_{GS}$ can be efficiently generated online via hashing of the token's global index, and its computation is identical to standard multinomial sampling in $\pi_{SJD}$. In Tab. 4, we present a latency breakdown of each component within single NFE step. As shown, the Transformer forward dominates the latency, and overhead of sampling operations, including our couplings, is under $\sim 5\%$

**Effect on CFG** AR vision models typically rely on CFG techniques to control prompt alignment and fidelity, using scale around 3~7. Specifically, the samples are generated from a mixed logit: $(1 + \lambda) \cdot c - \lambda \cdot u$, where logit $c$ generated with prompt and $u$ generated with masked prompt. As shown in Fig. 5, as $\lambda$ increases, speedup slightly decreased because the final logit becomes sharper. However, our method consistently outperforms SJD by large margin in practical range of scale $\lambda$.

Table 4: Latency breakdown ($ms$) per NFE step (Janus-Pro 7B, RTX 3090).

| Operation | $L=16$ | $L=32$ | $L=64$ |
|---|---|---|---|
| Preprocessing | 1.52 | 1.53 | 1.58 |
| **Transformer Fwd** | **26.49** | **27.73** | **36.41** |
| Logit Proc. | 0.16 | 0.23 | 0.25 |
| **Token Sampling (GS)** | **0.13** | **0.13** | **0.14** |
| **Vec. MRS (MC)** | **1.56** | **1.58** | **1.66** |
| Post Processing | 0.81 | 0.82 | 0.84 |

**Qualitative Results:** While we have theoretically and quantitatively demonstrated the lossless property of our method, we also performed a qualitative comparison experiment to visualize that our method does not degrade generation quality. As shown in Fig. 7, our method yields outputs that are visually indistinguishable from the AR model while achieving the $\sim 3\times$ acceleration. We provide more visualizations in the Appendix.

## 6 CONCLUSION

In this paper, we identify and resolve a critical performance bottleneck in the recently proposed Self-SD framework for autoregressive image generation, Speculative Jacobi Decoding (SJD). Specifically, we find that the speed potential of SJD is severely limited by its context instabilities, arising from an independent draft sampling process. To solve this problem, we propose to use an information-theory-inspired approach, Coupling, to replace the draft sampling and stabilize the Jacobi iteration trajectory by increasing the probability of re-sampling the token, transferring distributional similarity to the realized discrete token space. As a result, we show that this simple tweak can remarkably enhance the speedup of SJD, achieving $\sim 3.8\times$ to $\sim 10\times$ acceleration in visual AR generation, while maintaining its training-free and lossless properties.

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

APPENDIX

# A PROOFS

## A.1 PROOF OF PROPOSITION 1

*Proof.* We will first check that $\texttt{MRS}(\cdot)$ returns $Y \sim P$ with input $X \sim Q$. Let the acceptance probability $min(1, p(x)/q(x)) = \alpha(x)$. Then, we can re-write the p.d.f of R.V $Y$, $y(x)$ as follows

$$y(x) = \alpha(x) \cdot q(x) + (1 - \sum_{x' \in V} \alpha(x') \cdot q(x')) r(x) \tag{5}$$

where $r(x)$ is residual distribution $r(x) = norm(max(0, p(x) - q(x)) = \frac{max(0, p(x) - q(x))}{\sum_{x' \in V} max(0, p(x') - q(x'))}$.
We can rewrite the left term as :

$$\alpha(x) \cdot q(x) = min(1, p(x)/q(x)) \cdot q(x) = min(q(x), p(x)) \tag{6}$$

also with the right term :

$$(1 - \sum_{x' \in V} \alpha(x') \cdot q(x')) r(x) = (1 - \sum_{x' \in V} min(q(x'), p(x'))) \frac{max(0, p(x) - q(x))}{\sum_{x' \in V} max(0, p(x') - q(x'))} \tag{7}$$

$$= (1 - \sum_{x' \in V} min(q(x'), p(x'))) \frac{p(x) - min(p(x), q(x))}{\sum_{x' \in V} p(x') - min(p(x'), q(x'))} \tag{8}$$

$$= (1 - \sum_{x' \in V} min(q(x'), p(x'))) \frac{p(x) - min(p(x), q(x))}{1 - \sum_{x' \in V} min(p(x'), q(x'))} \tag{9}$$

$$= p(x) - min(p(x), q(x)). \tag{10}$$

So, adding two terms becomes $p(x)$, the target distribution, as desired.

Now, we will check the acceptance rate :

$$\mathbb{E}_{x' \sim q(x)} min(1, \frac{p(x')}{q(x')}) = \sum_{x' \in V} q(x') \cdot min(1, \frac{p(x')}{q(x')}) = \sum_{x' \in V} min(p(x'), q(x')) \tag{11}$$

$$= 1/2 \sum_{x' \in V} p(x') + q(x') - |p(x') - q(x')| = 1 - \frac{1}{2} \sum_{x' \in V} |p(x') - q(x')| \tag{12}$$

which is $1 - \mathcal{D}_{TV}(p, q)$.

## A.2 PROOF OF PROPOSITION 2

To compute value of $C_{SJD}$, let $p(x) = p_i^{(t)}(x)$ and $q(x) = p_i^{(t-1)}(x)$ for simplicity.

$$C_{SJD}(p, q) \equiv Pr[X^{(t)} = X^{(t-1)}] \tag{13}$$

$$= \sum_{x \in \mathcal{V}} Pr[X^{(t)} = x, X^{(t-1)} = x] \tag{14}$$

$$= \sum_{x \in \mathcal{V}} Pr[X^{(t)} = x] \cdot Pr[X^{(t-1)} = x] \qquad \text{(by independence)} \tag{15}$$

$$= \sum_{x \in \mathcal{V}} p(x) q(x) \tag{16}$$

Now, we will derive it's upper bound as follows :

$$(C_{SJD}(p,q))^2 = \left(\sum_x p(x)q(x)\right)^2 \tag{17}$$

$$\leq \left(\sum_x p(x)^2\right)\left(\sum_x q(x)^2\right) \qquad \text{(by Cauchy-Schwarz)} \tag{18}$$

Since Renyi-2 entropy is, by definition, $H_2(p) = -\log\left(\sum_x p(x)^2\right) \implies \sum_x p(x)^2 = e^{-H_2(p)}$,

Hence,

$$(C_{SJD}(p,q))^2 \leq e^{-H_2(p)} \cdot e^{-H_2(q)} = e^{-(H_2(p)+H_2(q))} \tag{19}$$

$$\implies C_{SJD}(p,q) \leq e^{-\frac{1}{2}(H_2(p)+H_2(q))} \tag{20}$$

So we can check that independence collision probability is exponentially restricted by their Renyi-2 entropy, regardless of how they are close to each other.

### A.3 Proof of Theorem 1

*Proof sketch.* The theoretical correctness of our approach is based on the marginalization property of the couplings. The standard SJD framework requires that the draft token, which we denote $X_i^{(t)}$, be sampled from the draft distribution $p_i^{(t)}$. If sample $X_i^{(t)}$ follows it's distribution, then correctness of SD framework is guaranteed by Proposition 1. According to **Definition 1**, when we sample a pair $(X_i^{(t)}, X_i^{(t-1)}) \sim \pi(\cdot,\cdot)$ from any valid coupling $\pi \in \Pi_i^{(t)}$, the marginal distribution of the variable $X_i^{(t)}$ is precisely $p_i^{(t)}$. Thus, using the $X_i^{(t)}$ component from the sampled pair is probabilistically identical to sampling a token directly from $p_i^{(t)}$. Since this modification preserves the required sampling distribution for the draft token at each step, the final output distribution of the algorithm is guaranteed to match that of the base model.

### A.4 Proof of Theorem 2

We will formally check that $\mathtt{MRS}(\cdot)$ satisfies the definition of Coupling. Let the joint distribution of this $\mathtt{MRS}(\cdot)$ process $f(x,y)$ is

$$f(x,y) = q(x)(\alpha(x)\delta_x(y) + (1-\alpha(x))r(y)) \tag{21}$$

where $\delta_x(y)$ is kronecker-delta symbol.

Let $\alpha(x)$, $r(x)$ is same as we defined on proof of proposition 1. Then for $p(y)$,

$$\sum_{x \in V} f(x,y) = a(y) \cdot q(y) + (1 - \sum_{x \in V}\alpha(x)q(x))r(y) = p(y) \tag{22}$$

which directly came out from proof of proposition 1.

Then next, for $q(x)$,

$$\sum_{y \in V} f(x,y) = q(x)\alpha(x)\sum_{y \in V}\delta_x(y) + q(x)(1-\alpha(x))\sum_{x \in V}r(y) \tag{23}$$

$$= q(x)\alpha(x) + q(x)(1-\alpha(x)) = q(x) \tag{24}$$

So it satisfies the definition of Coupling.

For the coupling cost optimality, it is well studied that any coupling can not have cost greater than $1 - \mathcal{D}_{TV}(P,Q)$ (Lindvall inequality) See Lindvall (2002); Bavarian et al. (2016).

## A.5 Proof of $\alpha$-Coupling

For Sec 5.1, we define $\alpha$-Coupling as follows :

$$\pi_{\text{cpl}}^{\alpha}(x,y) = \alpha \cdot \pi_{\text{cpl}}(x,y) + (1-\alpha) \cdot \pi_{\text{ind}}(x,y), \qquad (25)$$

wher $0 \leq \alpha \leq 1$. We will now show this linear-interpolaton of two coupling is still valid coupling.

To check marginal property, we can rewrite it as :

$$\sum_{y \in V} \pi_{cpl}^{a}(x,y) = \sum_{y \in V} [\alpha \cdot \pi_{cpl}(x,y) + (1-\alpha) \cdot \pi_{ind}(x,y)] \qquad (26)$$

$$= \alpha \cdot \sum_{y \in V} \pi_{cpl(x,y)} + (1-\alpha) \cdot \sum_{y \in V} \pi_{ind}(x,y) \qquad (27)$$

$$= \alpha \cdot p(x) + (1-\alpha) \cdot p(x) = p(x) \qquad (28)$$

Same procedure can be applied to $q(y)$.

**Implementation** To implement linear interpolation of two probability distribution, we can simply sample R.V from Bernoulli distribution, and pick sample with that probability. For example :

---

**Algorithm 5** $\alpha$-Coupling Implementation

---
**Input**: $\alpha$, Two valid coupling $\pi_a$, $\pi_b$.
**Output**: Random Variable X.

1: $u \sim \mathcal{U}(0,1)$
2: **if** $u \leq \alpha$ **then**
3: $\quad X, Y \leftarrow \pi_a(x,y)$
4: **else**
5: $\quad X, Y \leftarrow \pi_b(x,y)$
6: **end if**
7: **return** $X$

---

## B Related Works

**Unified Multimodal Models.** Recently, Unified Multimodal Models Team (2024); Deng et al. (2025); Hurst et al. (2024), which can process data from multiple modalities such as text, images, and audio for both input and output within a single model, have gained significant attention. The advantage of this paradigm stems from the discovery that models trained on multiple data domains simultaneously exhibit superior performance across a range of tasks compared to single-modality models. This includes enhanced understanding, generation Chen et al. (2025), complex world reasoning Hurst et al. (2024), instruction following, and iterative editing Bai et al. (2023).

**Autoregressive Models in Vision** Visual generation using an autoregressive (AR) Team (2024) approach is a promising method for implementing Unified Multimodal Models. An AR vision model primarily consists of two key components: a Vector Quantizer Van Den Oord et al. (2017) and a Transformer model Brown et al. (2020). The vector quantizer divides an image into patches of a specified size and maps each patch to a discrete code from a predefined codebook. This process effectively performs both downsampling and tokenization of the image. Subsequently, similar to autoregressive text generation, a Transformer model is trained to predict these visual token IDs autoregressively. This paradigm enables the learning and inference of diverse data types under a single, unified framework of AR modeling, naturally facilitating stable training, deployment, and capabilities such as in-context learning Hurst et al. (2024), editing Liu et al. (2024), and reasoning Zhao et al. (2025).

**Speculative Decoding** Speculative Decoding (SD) was first proposed by Leviathan et al. (2023); Chen et al. (2023) to accelerate the inference speed of Large Language Models (LLMs) without compromising performance by generating multiple tokens at once. Later, Sun et al. (2023) established a

connection between speculative sampling and optimal transport, proving that the token-level acceptance scheme is theoretically optimal for individual tokens. More recently, Sun et al. (2024b) showed that token-level acceptance is not globally optimal and that the block-wise acceptance approach is the theoretically optimal form of speculative decoding. As the theoretical optimality has been established, the recent research trend in SD has focused on designing better draft models **?**Brown et al. (2024); Cai et al. (2024) or exploring methods that trade speed for a slight degradation in quality Bachmann et al. (2025); So et al. (2025).

**Parallel Decoding** Parallel decoding, or fixed-point iteration $X \leftarrow F(X)$, is a widely used technique for rapidly finding the solution to a specific system, from scientific computing for accelerating the solution of differential equations Berinde (2004) to, more recently, fast sampling of diffusion models Shih et al. (2023). Building on this concept, Song et al. (2021) first proposed using fixed-point iteration to accelerate the sequential computation of neural networks. Based on the observation that this method guarantees the same result as sequential computation and always at least as faster than sequential when assuming fully parallelization model. Our method can be framed as a novel methodology for accelerating the convergence speed of fixed-point (jacobi) iteration for sequential sampling that operates based on a probabilistic process within a discrete space.

## C  LIMITATIONS

Since our method relies on parallel computation, the overhead of parallel operations may become significant if the window size or batch size becomes too large, potentially failing to achieve practical speedups, especially in scenarios where the parallel-computation unit is limited. However, this is a structural limitation shared by all Speculative Decoding (SD) methods, and we predict that these limitations will gradually disappear with the advancement of modern hardware.

## D  EXPERIMENTAL DETAILS

### D.1  IMAGE GENERATION

**Lumina mGPT:** For Lumina-mGPT Liu et al. (2024) , we use the standard 7B model and experiment with resolution of 768×768. In all experiments, we follow the default settings of vanilla model, temperature $\tau = 1$ and Top-K sampling with $K = 2000$ and guidance scale of $\lambda = 3.0$. We used pytorch 2.3 Paszke et al. (2019) for the main comparison. For quality evaluation, we generate 5000 images for each MS-COCO 2017 (val) Lin et al. (2014) prompt and compute FID, IS, CLIP-Score with reference dataset.

**Janus Pro :** For Janus-Pro Chen et al. (2025), we use 7B model to generate images at a resolution of $384 \times 384$. Following the setup of the vanilla Janus-Pro 7B model, $24 \times 24$ of image tokens are generated with a downsampling size of 16. For sampling, we follws vanilla setting that guidance scale of 5.0 and temperature of 1.0. We also adopted a Top-$K$ logits processor with $K = 1000$. For evaluation, we generate three images for each MS-COCO (val) prompt with different seeds (5000×3) and reported the mean values of the FID, IS, and CLIP score across the seeds.

### D.2  VIDEO GENERATION

**Cosmos1-autoregressive.** We evaluate our method on the *Cosmos-1.0-Autoregressive-4B* video AR model Agarwal et al. (2025) using a curated subset of 150 clips from the *real-state-10k* dataset Zhou et al. (2018). For each clip, we provide a 9-frame context to the model and autoregressively generate the next 24 frames, yielding 33-frame sequences in total (9 observed + 24 predicted). Unless otherwise noted, decoding uses nucleus (top-$p$) sampling with $p = 0.8$ and temperature 1.0.

We compare three decoders: (A) vanilla AR, (B) Speculative Jacobi Decoding (SJD), and (D) our MC-SJD on top of SJD. For SJD-based methods we sweep the parallel verification window $L \in \{16, 32, 64, 128\}$. Speed is reported as (i) **NFE**—the number of sequential target-model evaluations—and (ii) end-to-end wall-clock **Latency** (seconds) measured on a single RTX6000ADA. Quality is measured by **FVD** (lower is better), computed between the generated frames and the corresponding ground-truth future frames of each clip.

# E  ALGORITHMS

We provide complete pseudo code of our SJD with Gumbel Coupling in Algorithm 5.

---
**Algorithm 6** Pseudo Code for our **GS-SJD**

---
**Require:** AR Model $p_\theta$, draft Length $L$, Max Sequence $N$
1: $p_t^i \leftarrow \texttt{Random}()$; $X_i^t \sim p_i^t$                    ▷ Initialize state
2: **for** $j = 0$ to $N - 1$:                    ▷ Initialize shared Gumbel noise (Alg. 6)
3:    $G_j \leftarrow \texttt{SampleGumbelNoise}(|\mathcal{V}|)$

4: **while** $i < N$ **do**
5:    **parallel for** $j = i$ to $i + L$ :                    ▷ Drafting
6:       $X_{j,\_}^t \leftarrow \texttt{GS}(p_j^t, p_j^{t-1}, G_j)$

7:    **parallel for** $j = i$ to $i + L$ :                    ▷ Evaluate
8:       $p_j^{t+1} \leftarrow p_\theta(\cdot \mid X_{<j}^t)$

9:    **for** $j = i$ to $i + L$ :                    ▷ Verify
10:       $k, X_j^{t+1} \leftarrow \texttt{MRS}(p_j^{t+1}, p_j^t, X_j^t)$, **if** $k = 0$ : **break**

11:    $i \leftarrow j, t \leftarrow t + 1$
12: **end while**
13: **return** $X$

---

---
**Algorithm 7** $\texttt{SampleGumbelNoise}(V)$

---
**Input**: Vocabulary size $V = |\mathcal{V}|$.
**Output**: A Gumbel noise vector $G$ of size $V$.

1: $G \leftarrow [\,]$                    ▷ Initialize an empty list
2: **for** $i = 1 \rightarrow V$ **do**
3:    $u_i \sim \mathcal{U}(0, 1)$                    ▷ Sample from a standard uniform distribution
4:    $g_i \leftarrow -\log(-\log(u_i))$                    ▷ Apply inverse transform sampling
5:    Append $g_i$ to $G$
6: **end for**
7: **return** $G$

---

# F  MORE VISUALIZATION

In this section, we provide further details about the visualization settings and discuss our findings based on both quantitative and qualitative results. For image generation, we employed prompts covering diverse categories such as humans, animals, landscapes, close-up shots, fantasy, and paintings. In particular, we included prompts designed to capture physical phenomena such as reflections and waves. We also incorporated descriptors explicitly indicating high-quality imagery (e.g., 8K, sharp focus) to encourage the generation of fine-detailed, realistic images.

As shown in Figs. 9, 10, 11, 12, we observed that our method produced images closely resembling those of the vanilla AR model while achieved more than a $4\times$ reduction in NFE in image generation and $13\times$ in video generation. Moreover, our model was able to generate diverse categories of images, including physical phenomena like reflections and waves, under both the maximal coupling and the Gumbel coupling.

# G  MORE DATASET

To validate the generalization capabilities of our methods across various datasets, we conducted a new experiment using the Parti-Prompt Yu et al. (2022) dataset for text-to-image generation. Specifically, we evaluated generation quality using the CLIP score, and utilized a set of 1,600 distinct real-world text prompts for evaluation; all other experimental settings remain identical to the MS-COCO evaluation described in the main paper.

Table 5: Evaluation results of AR Image generation model, Lumina-mGPT, on Parti-Prompt.

| Configuration | | NFE (↓) | Latency (↓) | Acceleration (↑) | | CLIP-Score (↑) |
|---|---|---|---|---|---|---|
| | | | (A100) | NFE | Latency | |
| A | *Vanilla AR* | 2392 | 112.29 | 1.00× | 1.00× | 32.091 |
| B | SJD (L=16) | 1035.3 | 42.07s | 2.31x | 2.67x | 32.11 |
| D | + **Ours** ($\pi_{MC}$) | **817.29** | **32.86s** | **2.92x** | **3.42x** | 32.12 |
| D | + **Ours** ($\pi_{GS}$) | 820.12 | 32.92s | 2.91x | 3.41x | 32.07 |
| B | SJD (L=32) | 1038.2 | 43.27s | 2.30x | 2.60x | 32.087 |
| D | + **Ours** ($\pi_{MC}$) | **647.08** | **26.01s** | **3.69x** | **4.32x** | 32.09 |
| D | + **Ours** ($\pi_{GS}$) | 649.54 | 26.33s | 3.68x | 4.26x | 32.102 |
| B | SJD (L=64) | 1036.2 | 42.99s | 2.31x | 2.61x | 32.095 |
| D | + **Ours** ($\pi_{MC}$) | **548.26** | 23.58s | **4.36x** | 4.76x | 32.113 |
| D | + **Ours** ($\pi_{GS}$) | 548.75 | **23.42s** | **4.36x** | **4.79x** | 32.089 |
| C | GSD (G=50) | 636.75 | 25.24s | 3.76x | 4.45x | 32.075 |

| Config | | NFE (↓) | Latency(s)(↓) | CLIP-Score (↑) |
|---|---|---|---|---|
| A | *Vanilla AR* | 576 | 16.84s | 28.98 |
| B | SJD (L=16) | 312.00 | 9.96s | 29.02 |
| D | + **Ours**($\pi_{MC}$) | **186.48** | **6.22s** | 28.99 |
| D | + **Ours**($\pi_{GS}$) | 189.99 | 6.336 | 37.13 |
| B | SJD (L=32) | 308.17 | 10.25s | 28.97 |
| D | + **Ours**($\pi_{MC}$) | **149.53** | 5.471 | 38.34 |
| D | + **Ours**($\pi_{GS}$) | 154.42 | **5.388** | 37.49 |

Table 6: Evaluation results of AR Image generation model, Janus-Pro, on Parti-Prompt.

Tables 5 and 6 present the results on the Parti-Prompt dataset. As shown, our methods exhibit acceleration rates of approximately $\sim 4\times$, which are comparable to those observed on the MS-COCO dataset, demonstrating the strong generalization capabilities of our proposed methods.

# H   USE OF LLM

We used a Large Language Model (LLM) for typo checking, grammar correction, and polishing of our paper draft.

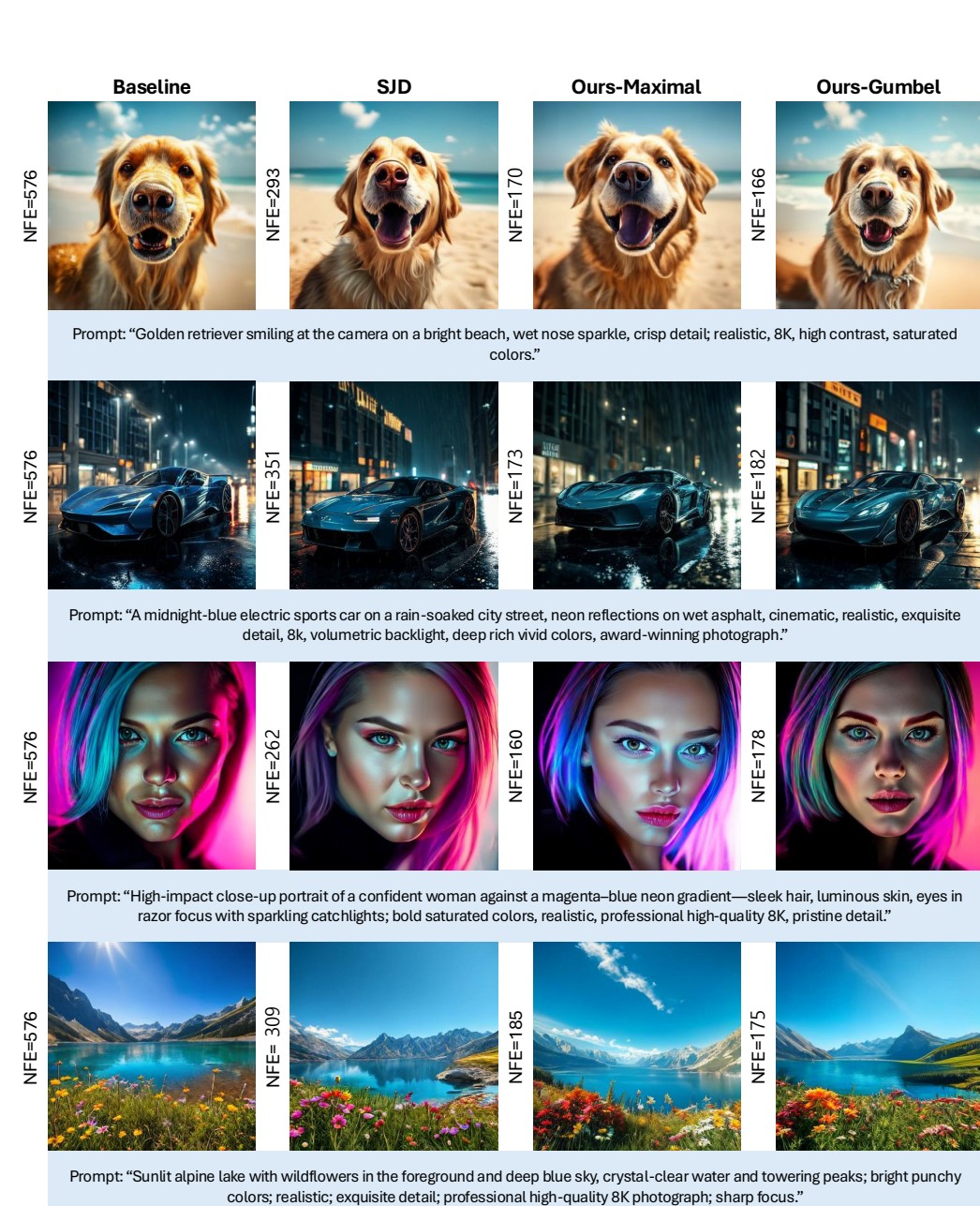

Figure 9: Qualitative comparison on Janus-Pro 7B

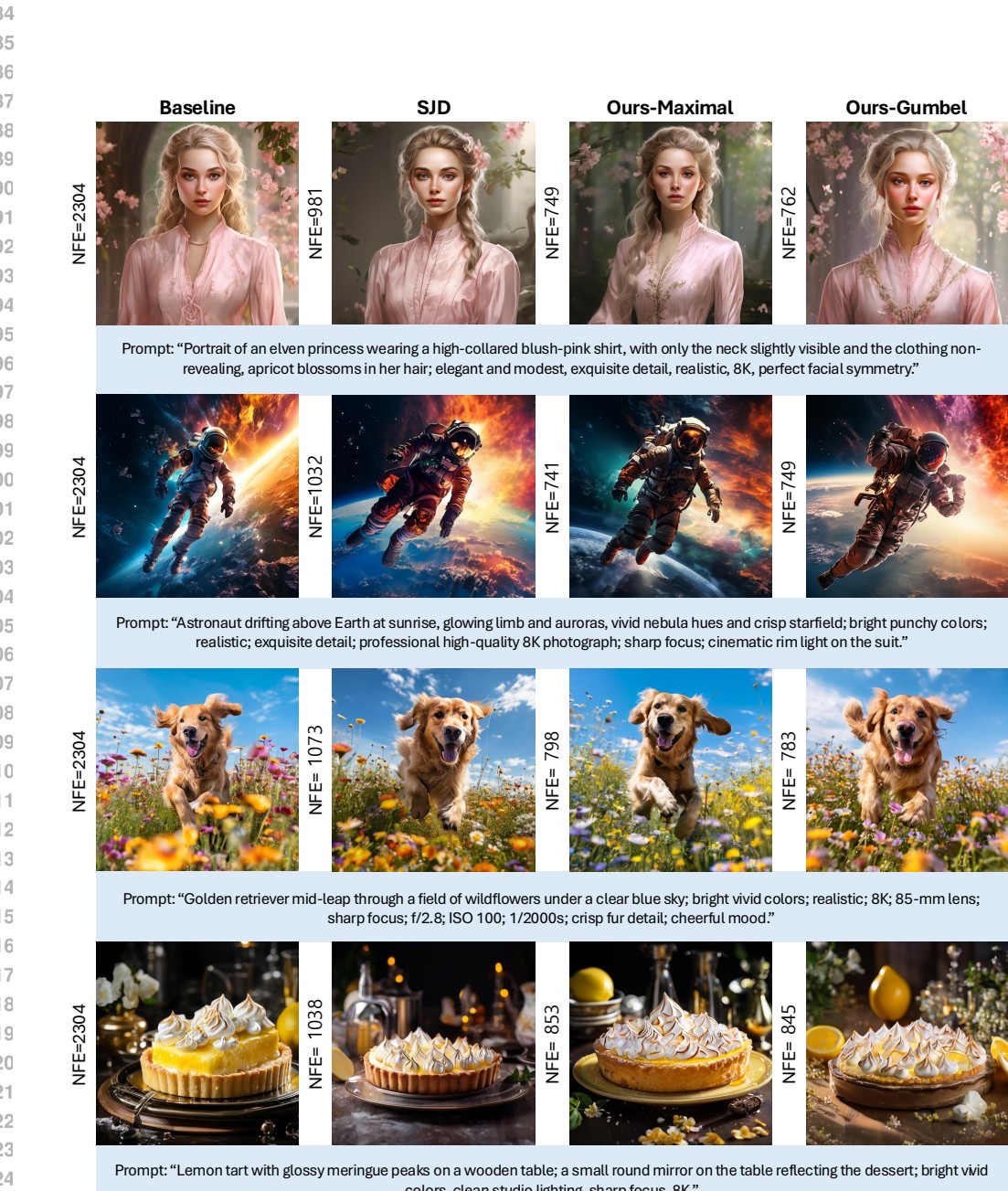

Figure 10: Qualitative comparison on Lumina-mGPT (1.0)

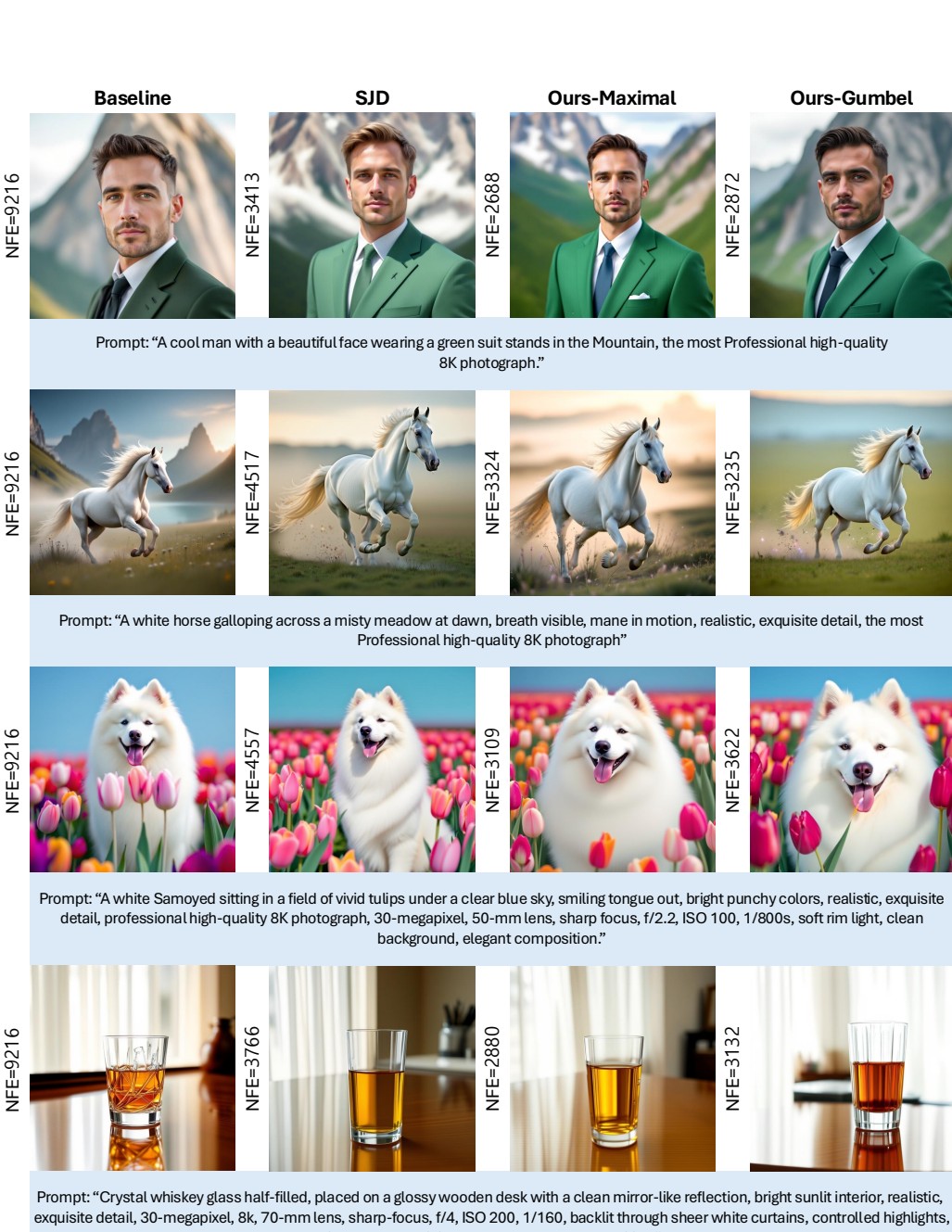

Figure 11: Qualitative comparison on Lumina-mGPT 2.0

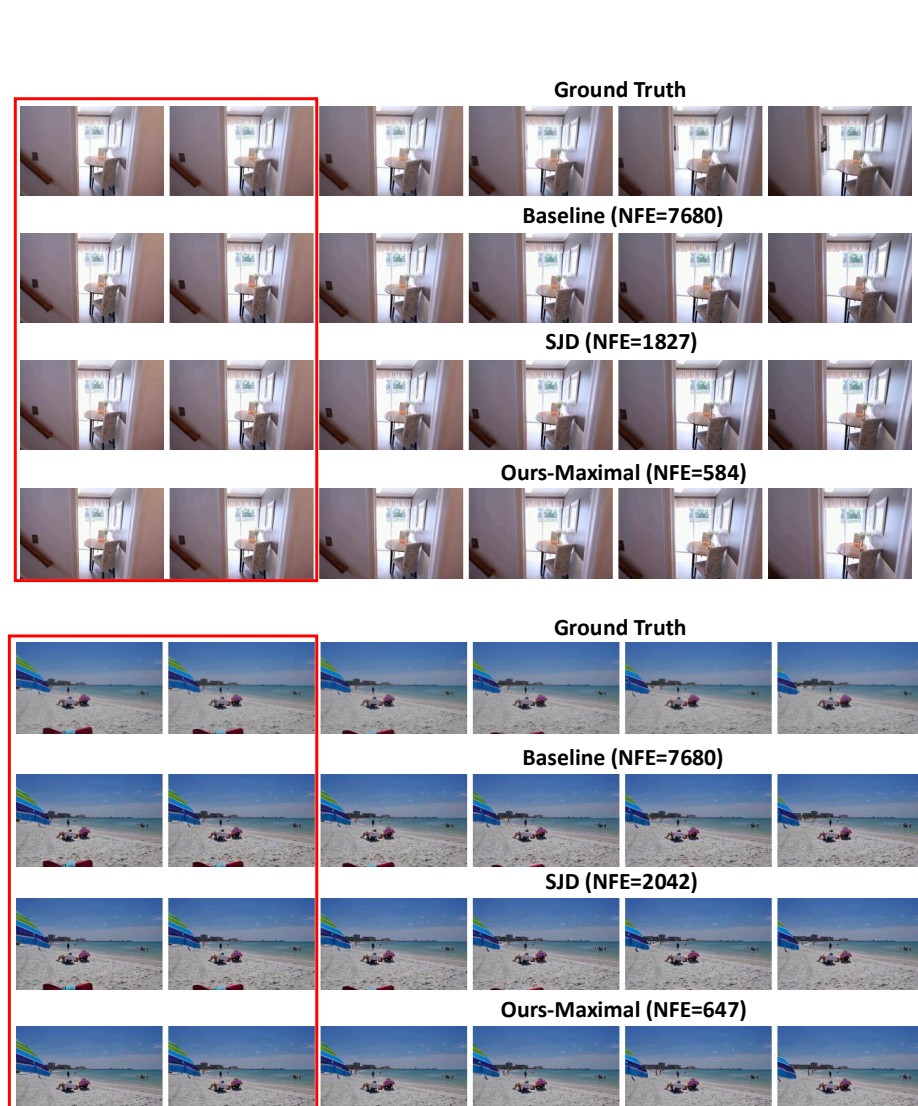

Figure 12: Qualitative comparison on Video Generation ( Cosmos-1-ar )

