# OpenReview forum: "MC-SJD : Maximal Coupling Speculative Jacobi Decoding for Autoregressive Visual Generation Acceleration"
_ICLR.cc/2026/Conference — Submitted to ICLR 2026_

### Official Review · Reviewer_cQ2e · 2025-11-01

**Soundness:** 3
**Presentation:** 2
**Contribution:** 3
**Rating:** 4
**Confidence:** 4

**Summary:**

This paper addresses the slow inference of autoregressive (AR) visual models by improving Speculative Jacobi Decoding (SJD). The authors identify that SJD's performance is bottlenecked by token instability, where independent sampling creates dissimilar draft tokens even when underlying probability distributions are close. The proposed solution, MC-SJD, is a lossless and training-free framework that uses "Coupling" to maximize the token similarity between iterations, thereby stabilizing convergence. The method demonstrates impressive speedups, achieving up to ~$3.8\times$ in image and ~$10\times$ in video generation without any degradation in output quality.

**Strengths:**

* Well-written and easy to follow.
* Provides a novel problem and method in SJD based on the theoretically sound principle of coupling.
* Demonstrates significant empirical speedups over baselines. The inclusion of experiments on both image and video generation tasks is a positive aspect of the evaluation.

**Weaknesses:**

1. **Limited Empirical Support for Motivation**
	* The correlation in Fig. 2, which is used to empirically validate Observation 1, is not fully convincing. The Y-axis is restricted to an extremely narrow range (59.0–61.0), which makes the "strong correlation" difficult to confirm. The argument would be far more compelling if SJD, GS-SJD, and MC-SJD were all plotted on the same graph. This direct comparison would clearly visualize whether the proposed methods truly shift performance toward lower token difference and lower NFE, which is central to the paper's hypothesis.

2. **Apparent Discrepancy Between Theory and Results**
	* An interesting discrepancy arises in Table 1 ($L=32$), where the theoretically suboptimal $\pi_{GS}$ (Gumbel Coupling) achieves a better NFE than the "optimal" $\pi_{MC}$. The paper fails to provide any analysis for this critical discrepancy, undermining its own central argument.

3. **Incomplete Experimental Reporting**
	* Tables 2 and 3 report results for an ambiguous "Ours" metric, failing to specify whether $\pi_{MC}$ or $\pi_{GS}$ was used. As established in Weakness 2 (and Table 1), the performance trade-off between $\pi_{MC}$ and $\pi_{GS}$ seems inconsistent and setup-dependent. Therefore, it would be necessary to report results for both methods to provide a complete and transparent comparison.
	* Table 3 (Janus-Pro) omits wall-clock latency. This is a critical metric for evaluating the method's practical benefit, especially given the computational overhead of $\pi_{MC}$ and $\pi_{GS}$ observed in Table 1.
	* The validation for each task is confined to a single dataset (MSCOCO2017 for images, real-state-10k for video), which limits the assessment of the method's generalizability.

4. **Lack of Limitation**
	* The paper does not provide any discussions regarding limitations.

**Questions:**

1. The paper speculates that Gumbel Coupling ($\pi_{GS}$) promotes "long-range stabilization". Could the authors elaborate on this concept? Specifically, what is the mechanism of this stabilization (e.g., how does sharing the same Gumbel noise across all iterations achieve this effect), and what is its concrete role in the decoding process? Does it, for example, help the entire sequence trajectory converge to a stable state more quickly?

2. Following the first question and weakness 2, Table 1 shows that at $L=32$, $\pi_{GS}$ achieves a better NFE than the theoretically optimal $\pi_{MC}$. Does this finding suggest that the "greedy" optimization of $\pi_{MC}$ (maximizing immediate $t$ vs. $t-1$ collision) is not always the globally optimal strategy for minimizing total NFE? Does it imply that, depending on the setup (like window size L, model, or task, etc), the "long-range" approach of $\pi_{GS}$ can actually be the superior strategy?

---

> ### Author Response · Authors · 2025-11-21
> **Response to cQ2e (1/4)**
>
> Thank you for the thoughtful reviews and for pointing out our paper's novelty and significant performance improvements. Below, we address each concern in detail.
>
> ## Weakness 1: Additional Empirical Data for Coupling
>
> Thank you for the insightful suggestion. To address your request, we have **conducted new experiments and added a new Section 5.1.1** to discuss these results. Specifically, to visualize SJD and MC-SJD on a single plot and provide a more intuitive understanding of the coupling effect, we define **$\alpha$-Coupling** as a linear interpolation between the independent distribution $\pi_{\text{ind}}$ of SJD and our coupling distribution $\pi_{\text{cpl}}$ (using $\pi_{\text{MC}}$ or $\pi_{\text{GS}}$), controlled by a factor $\alpha$. We then show the changes in Token Difference and NFE according to $\alpha$ (Extension of Fig. 2). Specifically, we define $\alpha$-Coupling as:
>
> $\pi_\alpha = \alpha \cdot \pi_{\text{cpl}} + (1-\alpha) \cdot \pi_{\text{ind}}$
>
> Here, $\alpha=0$ corresponds to the existing SJD (Fig. 2), and $\alpha=1$ corresponds to Ours. A proof of $\alpha$-Coupling's validity and implementation is included in Appendix A.5.
>
> The results are presented in Fig.6 and Section 5.1.1. As we make the *Coupling Strength* $\alpha$ higher, both Token Difference and NFE decrease. This shows that context difference is actually highly correlated with the acceptance rate, and that reducing this token difference by increasing the collision probability significantly helps in reducing total NFEs. Furthermore, this trend continues up to $\alpha \approx 1$, showing that coupling the two consecutive draft sequences as close to their theoretical upper bound as possible yields near-optimal results.

---

> ### Author Response · Authors · 2025-11-21
> **Response to cQ2e (2/4)**
>
> ## Weakness 2, Question 1, Question 2: Optimality and Difference of MC and GS
>
> Since these three points are deeply connected with the same issue—the difference between $\pi_{MC}$ and $\pi_{GS}$—we will answer them together.
>
> **Optimality of $\pi_{MC}$**: First, we want to clarify that our claim of $\pi_{MC}$ being "theoretically optimal" is limited to the perspective of **collision probability within a single 1-step** $(t, t-1)$ throughout the paper. We did not claim **global optimality** regarding the total NFE, which is ultimately determined by highly complex non-linear dynamics of Transformer model. Analyzing this global behavior in closed-form is extremely difficult, and we tried to focus on analyzing and improving the theoretically available local, 1-step behavior of SJD. Nevertheless, throughout the paper and in the additional experiments (W1, Sec. 5.1), we have empirically shown that this local 1-step behavior is strongly correlated with global convergence and final NFE, and maximizing their 1-step collision probability as close to their theoretical upper bound as possible yields near-optimal results. We have revised Sec. 4.2 to clarify this more clearly.
>
> **Difference between $\pi_{MC}$ and $\pi_{GS}$**: While the locally slightly sub-optimal $\pi_{GS}$ can surely have slightly better global results because there is no "strict" correlation between local behavior and global behavior, **we have added a new discussion in Section 5.1.2** to provide a deeper analysis of the case where $\pi_{GS}$ yields a slightly better NFE than $\pi_{MC}$. As shown in the new Fig. 8, $\pi_{GS}$ has a higher Hamming distance (token difference) than $\pi_{MC}$ in a 1-step context, which is aligned with our paper's main theory. However, in $N$-Step ($N>1$) cases, $\pi_{GS}$ results in a significantly lower Hamming distance than $\pi_{MC}$, especially in the high coupling strength regime. This is due to the nature of Gumbel Coupling; as mentioned in Theorem 3, the lower bound of $\pi_{GS}$ coupling is guaranteed to be $(1-TV)/ (1+TV)$. Therefore, unless the Total Variation (TV) differs drastically between iterations, Gumbel Coupling forms a similar lower bound not only between "current consecutive iterations" (t, t+1) but also "future" iterations (t, t+N). In contrast, while $\pi_{MC}$ can quantify the exact coupling cost between (t, t+1), it is impossible to define a non-trivial lower bound for coupling costs between other iterations (t, t+N). Consequently, while MC ensures tokens are more similar on average within 1-step, they may become less similar on average over $N$-steps, and we actually observed this phenomenon in Figure 8. In this regard, we describe **MC as performing "Greedy Optimization"** on token similarity between iterations, whereas **GS offers "Long-range Stabilization."**
>
> However, as mentioned earlier, analyzing exactly how these local behaviors affect overall convergence and final NFE is fundamentally difficult due to the complex dynamics of non-linear Transformers, and it depends on the empirical characteristics of the task, model, or dataset. For instance, assuming a model where the initial iteration's Draft guess is relatively easy, maintaining the initial draft ($\pi_{GS}$) to increase overall convergence stability may lead to fewer NFEs than updating all tokens with slightly improved information in 1-step ($\pi_{MC}$). For example, for the Video AR in Table 3, the guessing difficulty is relatively low because this task involves only slightly changing the given initial frame; hence, we speculate that $\pi_{GS}$ yields fewer NFEs in i2v Video AR. Conversely, for Lumina (Tab 1), a relatively difficult text-to-image and high-resolution generation model, this trend is not distinct, and both couplings yield comparable NFEs. Regardless, both couplings provide significant improvements over the independent sampling of existing SJD. Therefore, rather than claiming one coupling is definitively better, we leave this as a choice for the user.

---

> ### Author Response · Authors · 2025-11-21
> **Response to cQ2e (3/4)**
>
> ## Weakness 3: Lack of Experiments
>
> * **Updated Tables:** We have updated Tables 2 and 3 to include values for both couplings. As shown, $\pi_{GS}$ have comparable or slightly better NFEs than $\pi_{MC}$ in Tab 2.3, and we speculate this depends on task difficulty; i.e., Janus can be easier than Lumina because it generates lower-resolution outputs, and i2V video AR can also be easier due to the presence of the initial frame condition. Nevertheless, our coupling methods introduce significant acceleration compared to existing AR and SJD in either case.
> * **Latency:** We have updated Table 3 to include **wall-clock latency**, clarifying that our method shows significant wall-time acceleration compared to other methods in Janus-Pro. Moreover, our coupler's overhead is negligible (~5%). Please see the global respoonse.
> * **Generalizability:** We added new experiments on the **Parti-Prompt dataset** for image generation experiments. The results are in new Tab.5,6. As shown, our methods still show significant acceleration on the new dataset, demonstrating the generalization capability of our methodology.
>
> ## Weakness 4 : Lack of Limitations
>
> We have added a new limitation section to the Appendix. Basically, since our method relies on parallel computation, the overhead of parallel operations may become significant if the window size or batch size becomes too large, potentially failing to achieve practical speedups, especially in scenarios where the parallel-computation unit is limited. However, this is a structural limitation shared by all Speculative Decoding (SD) methods, and we predict that these limitations will gradually disappear with the advancement of modern hardware.
>
> In contrast, our method requires no additional training or model modification, preserves model accuracy strictly, has negligible inference overhead, and has demonstrated generalization performance across various large-scale visual AR models including images and videos. Moreover, despite these advantages, we have achieved SOTA performance with a significant margin compared to current Visual AR SD standards.

---

> ### Author Response · Authors · 2025-11-21
> **Response to cQ2e (4/4)**
>
> Besides, we would like to thank the reviewer once again for the insightful review. The experiments and questions you suggested were all natural and important points, and we genuinely appreciate that they helped significantly improve the quality of our paper. However, $\pi_{MC}$ and $\pi_{GS}$ are both variants of our methodology that implement the core of our theoretical result, and the fact that two different coupling implementations show similar performance and significantly outperform existing methods is **actually a strength that serves as empirical evidence supporting our claim and theoretical foundation**. A granular comparison between the slight differences of these two variants is, of course, interesting future work, but it is a bit outside the scope of this paper, and we believe the lack of this is not a major weakness. We would be grateful if you could consider this perspective.

---

### Official Review · Reviewer_znnZ · 2025-11-01

**Soundness:** 4
**Presentation:** 4
**Contribution:** 3
**Rating:** 6
**Confidence:** 2

**Summary:**

The authors show that the bottleneck in Speculative Jacobi Decoding (SJD), low acceptance due to independent draft sampling, can be removed by coupling the draft distributions across iterations. They propose two couplers: Maximal Coupling (with acceptance equal to 1−TV(p,q)) and a cheaper Gumbel-Coupling (shared Gumbel noise) with a provable lower bound on collision probability; both plug into SJD with a one-line change in the drafting step. Theoretical analysis connects acceptance to total-variation distance and shows that independent sampling yields low collision probability bounded by Rényi-2 entropy, especially flat in visual AR; coupled sampling markedly raises acceptance trajectories βₜ (Fig. 3–4).

**Strengths:**

* Theoretical observation is very interesting. The paper replaces independent drafting by a principled coupling with formal guarantees (Theorem 2/3). The relationship between the acceptance and the total-variation is clear.

* Drop‑in practicality. The “single‑line” modification to SJD is attractive for practitioners and retains lossless correctness of speculative decoding.

* Strong speedups across both image and video domains, scaling favorably with larger SJD windows where vanilla SJD saturates.

**Weaknesses:**

I'm not an expert in this area, but I have some concerns about the paper based on my understanding.

1. I have some doubt on compute/memory cost of coupling. What is the runtime and memory overhead for maximal vs. Gumbel coupling (per step, per window L)? Any GPU kernel implications vs. independent sampling?

2. Is there any failure modes observed empirically? For example, when p and q are both flat (I think it would be common in AR images), TV is small but entropy is high. Does coupling still help, or do we hit the Rényi‑2 bound and stagnate? Please show acceptance curves for extreme‑flat logits.

3. CFG interaction: you note speed dips at higher guidance (sharper logits). Can you quantify the acceptance–guidance trade‑off and provide an adaptive lambda schedule for maximum throughput?

**Questions:**

Please refer to the weaknesses.

---

> ### Author Response · Authors · 2025-11-24
>
> Thank you for the positive assessment. In particular, we appreciate you highlighting the theoretical novelty, contribution, and strong performance improvements. Below is a discussion regarding your concerns.
>
> ## Weakness 1: Coupler Overhead
>
> Please refer to the global response. In actual implementation, our coupler uses the same operations already performed in SJD, so there is almost no additional overhead. Additionally, the overhead w.r.t. window size is negligible because they can be fully vectorized.
>
> ## Weakness 2: Flat distribution failure mode
>
> **Failure modes:** Although it is outside the scope of this paper, we have observed that MC-SJD does not work well in the text domain, **where the distribution is nearly one-hot**. In this case, the room for randomness in stochastic operations of MC-SJD—from SD accept to draft token coupling—disappears, converting entire decoding process into a static decoding process, JD [1]. However, extending to other data domains is beyond the scope, and at least within the domain of Visual AR (images, videos), we have not observed specific failure modes regarding the mentioned problem. In fact, since these flat-distribution characteristics mainly stem from the nature of Vector-Quantizer-based tokenizers, we believe this rather demonstrates the potential for extension to other AR domains that also use VQ tokenizers (e.g., Audio, 3D).
>
> **Flat Distribution:** In cases where TV is small but entropy is high, coupling surely helps. As explained in the paper, when using the proposed Coupling, the relationship between Collision Probability and Entropy disappears, and Collision Probability becomes a **function of TV**. Therefore, in MC-SJD, **flatness or Renyi Entropy does not directly influence Collision Probability**, and high values typically do not cause stagnation. The only influencing factor is the TV, and this has no direct trivial relationship with entropy.
>
> **Acceptance Curves for Extreme‑Flat Logits:** In the extreme case where two distributions are completely flat, they imply two identical uniform distributions, so the TV would be 0 and the acceptance rate would be 1. However, if the state is "slightly" off from perfectly uniform, the TV will be determined by other factors, such as vocab size or the "pattern" of differences. **This has no direct trivial relationship with entropy and ultimately depends on the model's behavior.** Therefore, rather than looking at acceptance curves relative to entropy, we believe it is more important to look at the TV observed empirically in actual tasks, as shown in Fig. 4.
>
> We hope this explanation helps alleviate your concerns. Please let us know if you would like any additional experiments.
>
> ## Weakness 3 : Guidance scale effects
> Thank you for the comment. In the table below, we present the NFE and Accel. ratio of AR, SJD, MC-SJD according to the guidance scale to quantify acceptance–guidance trade‑off . We used Lumina-1 and reported the avg. NFE values over 100 prompts.
>
> | | MCSJD | SJD | AR/MCSJD | SJD/MCSJD |
> | :--- | :--- | :--- | :--- | :--- |
> | **0.1** | 496 | 913 | 4.81x | 1.84x |
> | **1** | 577 | 949 | 4.14x | 1.64x |
> | **3** | 638 | 1027 | 3.74x | 1.61x |
> | **5** | 747 | 1097 | 3.19x | 1.47x |
> | **7** | 826 | 1173 | 2.89x | 1.42x |
> | **9** | 922 | 1270 | 2.59x | 1.38x |
>
> As seen in the table, as guidance increases, the gain of MC-SJD relative to AR and SJD diminishes. However, our method demonstrates acceleration at least superior to existing SJD across all guidance scales and shows good acceleration rates within reasonable range used in Image AR (3–5). Generally, extremely high guidance scales of 9 or above are rarely used.
>
> **Adaptive Lambda Schedule:** Thank you for suggesting an interesting experiment. Our idea for maximum throughput is as follows: in the later stages of token generation, since the image structures have already been generated, the impact of CFG on image generation can be minimal. Therefore, by using a lower $\lambda$ in the later stages of generation, we can expect greater acceleration while maintaining image quality.
>
> Below are the results; specifically, we used the guidance scale $\lambda=3$ for 75% of the iterations, and $\lambda=1$ for the final 25%. Similarly, we reported the average NFE values for 100 prompts using Lumina-1.
>
> | | NFE | CLIP-Score |
> | :--- | :--- | :--- |
> | AR | 2390 | 31.88 |
> | SJD| 1027 | 31.85 |
> | MC-SJD($\lambda=3$, static) | 633 | 31.94 |
> | MC-SJD($\lambda=3 \to 1$, schedule) | 605 | 31.90 |
>
> As shown, applying lambda scheduling provides a slightly lower NFE while maintaining the CLIP score. We think this is interesting future work, but please consider that the main objective of this paper is to increase decoding speed while *exactly* preserving existing AR performance, so we treat the guidance scale as a given constant rather than a tunable hyperparameter.
>
> [1]  Song, Yang, et al. "Accelerating feedforward computation via parallel nonlinear equation solving." ICML, 2021.

---

### Official Review · Reviewer_FLTy · 2025-11-02

**Soundness:** 3
**Presentation:** 3
**Contribution:** 3
**Rating:** 4
**Confidence:** 4

**Summary:**

The manuscript proposes MC-SJD, a training-free modification of Speculative Jacobi Decoding (SJD) for autoregressive (AR) visual generation. The manuscript analyzes that the acceptance rate is negatively proportional to the total variation of the drafter and verifier distributions, and argues that independent drafting in SJD yields very low collision under flat vision logits. Aware of this, they propose a key idea to couple the draft sampling across consecutive Jacobi iterations via maximal coupling (using the same MRS routine as SD verification) or a cheaper Gumbel noise–sharing variant to increase token collisions between drafts, thereby boosting acceptance rates while remaining lossless. Empirically, they report up to ~3.8× image speedup and ~10× video speedup without quality degradation.

**Strengths:**

1. Clean and intuitive problem statement - points out an overlooked but crucial detriment of SJD.
2. MC-SJD can be employed with minimal implementation change.

**Weaknesses:**

1. Baseline coverage is a bit limited. While SJD and one baseline (GSD) are included, there’s no comparison with other accepted speculative decoding baselines, such as EAGLE-3.
2. Latency analyses w.r.t. the batch size are absent.
3. Memory overhead analysis of the cached probabilities is needed, especially w.r.t. the batch size and window length.
4. Latency breakdown or microbenchmark for the sampling process would benefit the paper.

**Questions:**

1. When using top-k sampling (as in the experiments section) or using CFG, how is the lossless-ness guaranteed? How does the sampling process go?

---

> ### Author Response · Authors · 2025-11-21
> **Response for FLTy**
>
> We thank the reviewer for the valuable feedback.
>
> **Weakness 1: Comparison with other SD baselines (e.g., EAGLE)**
>
> The primary reason we compared our method only with SJD and GSD is that, to the best of our knowledge, these are only works on **Training-Free Speculative Decoding for Image AR**. While methods like EAGLE-2 can also apply SD to Image AR, they are **draft-model-based**, requiring the training and inference of a separate new draft model; thus, they are not plug-and-play applicable, and we consider them outside the scope of our direct comparison.
>
> However, following your request, we conducted comparative experiments with additional baselines, **EAGLE2 and LANTERN** [1], which are draft-model-based SD methods. We performed the experiment on **LlamaGen XL - Stage 2** using 500 MS-COCO prompts. The results (average NFE) are as follows:
>
> | Method | Training-Free? | Lossless? | NFE ($\downarrow$) |
> | :--- | :---: | :---: | :---: |
> | Standard AR | - | - | 1024. |
> | EAGLE2 | ✗ | ✓ | 853.30 |
> | LANTERN ($\delta = 0.1$) | ✗ | ✗ | 585.14 |
> | SJD | ✓ | ✓ | 632.49 |
> | **MC-SJD ($\pi_{MC}$)** | **✓** | **✓** | **515.72** |
>
> As shown, our MC-SJD surpasses all baselines by a large margin, outperforming the draft-model-based EAGLE and even the lossy-SD(LANTERN), while maintaining training-free and lossless-ness constraints. We will add a table comparing these existing SD methods with ours in the appendix.
>
> **Weakness 2: Experiments on Batched Inference**
>
> Thank you for the suggestion. However, we intentionally report results only with batch size = 1, following all prior speculative decoding works for visual AR [1, 2, 3]. Actually, using a batch size $>1$ in SD is practically hard to implement and non-trivial. Since each instance in a batch accepts a different number of tokens per iteration, one must either min-cut all sequences to the shortest accepted length or pad to the longest. The former can severely reduce the effective speedup, while the latter requires complex KV-cache re-packing and can introduce alignment/accuracy issues. For this reason, most SD papers restrict evaluation to batch = 1, and efficient batched SD is emerging as a separate research topic [4, 5].
>
> Moreover, while it is true that increasing batch size pushes operations toward a compute-bound regime—thereby reducing effective speedup SD gains—this is a structural limitation shared by **all SD paradigms**, not specific to our method. We have added this limitation to the Appendix.
>
> **Weakness 3: Memory Overhead of the Coupler**
>
> Thank you for pointing this out. Our coupler introduces **almost no memory overhead** compared to standard SJD, and there is no cached-probabilities in the real implementation. Please see the global response.
>
> **Weakness 4: Latency Breakdown**
>
> Thank you for the suggestion. Our coupler accounts for less than 5% of the total latency. Please see the global response.
>
> **Question 1: CFG, Top-K, and Losslessness**
>
> Thank you for the question. Applying CFG or Top-K sampling **does not affect the losslessness** of our method. This is because the target distribution $p(x)$ we aim to recover is defined as the **final distribution after all logit processing** (including CFG and Top-K) has been applied. Our algorithm is designed to sample exactly from this final modified distribution. We will clarify this definition in the final version of the paper to avoid ambiguity.
>
> [1] Jang, Doohyuk, et al. "Lantern: Accelerating visual autoregressive models with relaxed speculative decoding." ICLR'25.
> [2] Teng, Yao, et al. "Accelerating auto-regressive text-to-image generation with training-free speculative jacobi decoding." ICLR'25.
> [3] So, Junhyuk, et al. "Grouped speculative decoding for autoregressive image generation." Proceedings of the IEEE/CVF International Conference on Computer Vision. 2025.
> [4] Zhang, Ranran Haoran, et al. "Batch Speculative Decoding Done Right." arXiv preprint arXiv:2510.22876 (2025).
> [5] Qian, Haifeng, et al. "BASS: Batched attention-optimized speculative sampling." arXiv preprint arXiv:2404.15778 (2024).

---

### Author Response · Authors · 2025-11-21
**Global Response : Coupling Overhead**

# Overhead of Coupling



We thank the reviewers for the constructive feedback. In fact, our coupler introduces **almost no overhead** compared to standard SJD in the real implementation, and this property is actually one of the **key strengths** of our method. This is because of the following:

* **For $\pi_{MC}$:** Algorithm 2 in the paper is pseudo-code for a conceptually clear comparison with SJD, and there are actually no cached probabilities in the real implementation. In an actual efficient implementation, the for-loop in Line'5(in Alg.2) does not exist, and **we simply execute the verification loop in Line'9 without a break, storing the index of the first $k=0$ signal**. This can be easily validated by the fact that Line'5 and Line'9 are strictly identical operations because $p^{(t+1)}, p^{(t)}$ in Line'9 becomes $p^{(t)}, p^{(t-1)}$ in the next loop's Line'5. Thus, there is no additional memory used compared to standard SJD. Moreover, because we do not have to break Line'8 loop sequentially, we can **vectorize** this loop; thus, the computational overhead of MRS relative to the window Length is also negligible. We also show real latency breakdown of vectorized MRS in below.

* **For $\pi_{GS}$:** In an actual efficient implementation, we do not need to pre-sample and store Gumbel noise in memory. We can just sample Gumbel noise **online** by hashing the global index(which is only 4Bytes) of the current token to generate the Uniform RNG, eliminating memory overhead; the computational overhead of this online RNG sampling is negligible. Moreover, because the 'token sampling' part with Gumbel noise is computationally identical to the standard multinomial sampling used in independent sampling, GS also has negligible overhead in reality.

# Latency Breakdown

To also validate that our coupler has negligible overhead in practice, we implemented the efficient version of our coupler and present the wall-clock latency breakdown of each component in a single NFE step.

| Operation  | $L=16$ | $L=32$ | $L=64$ |
| :--- | :---: | :---: | :---: |
| Input Preprocessing | 1.52 | 1.53 | 1.58 |
| **Transformer Forward** | **26.49** | **27.73** | **36.41** |
| Logit Processing (CFG, Top-K) | 0.16 | 0.23 | 0.25 |
| **Token Sampling (GS-Coupling)** | 0.13 | 0.13 | 0.14 |
| **Vectorized MRS (MC-Coupling)** | 1.56 | 1.58 | 1.66 |
| Post Process | 0.81 | 0.82 | 0.84 |
(milliseconds, Janus-Pro, RTX 3090)


As shown, the Transformer forward pass dominates the total latency, and the overhead introduced by our coupling (Vectorized MRS or Token Sampling) is under $\sim$5\%, and even decreases when the window size increases. Moreover, we updated latency evaluation on our main tables(Tab.1,2,3) accordingly.

---

We will add difference between the logical representation of the algorithm in the paper and the actual implementation. Also, we will clarify that there is almost no overhead with our coupling with breakdown results. Besides, we also note that our use of Gumbel Coupling is driven not just by efficiency, but by its implementation simplicity (friendly for limited hardware like NPUs) and its positive impact on NFE at some cases.

---

### Author Response · Authors · 2025-11-26
**Reminder: Encouraging Rebuttal Discussion**

Dear Reviewers,

We would like to express our sincere gratitude for your thoughtful reviews, which have been invaluable in clarifying our contributions and strengthening our claims.

**As the discussion period concludes in one week, we kindly ask reviewers to review our responses and post a feedback.** We have made every effort to address your concerns, and we would be grateful if you would consider updating your evaluation to reflect these revisions.

---
We have also updated our manuscript to incorporate your feedback. All revisions are highlighted in *blue*. Key updates include:

* **Coupling Overhead:** In **Sec. 5.2**, we provided a detailed explanation of the efficient implementation of our coupler and included a latency breakdown analysis. These results demonstrate that the computational overhead is negligible in practice.
* **$\alpha$-Coupling Experiments:** We added a new **Figure 6** and corresponding descriptions in **Sec. 5.1** to provide a more intuitive comparison between SJD, MC-SJD, and GS-SJD and strengthen our core motivation.
* **Long-range Stability of $\pi_{GS}$:** We included experimental results (**Fig. 8**) and analysis regarding the long-range stability of $\pi_{GS}$ in **Sec. 5.1**.
* **Revision of Sec. 4.2:** We clarified potential misunderstandings regarding the optimality of $\pi_{MC}$ and explicitly stated that $\pi_{GS}$ may yield better performance in some scenarios.
* **More Experimental Results:**
    * We now report the performance of both couplers ($\pi_{MC}$ and $\pi_{GS}$) in **Main Tables 1, 2, and 3**.
    * We add additional results on the **Parti-Prompt** dataset in **Tables 5 and 6** at Appendix G.
* **Limitations:** We added a discussion on limitations in the **Appendix**.

---

### Author Response · Authors · 2025-12-04
**Rebuttal Summary for AC**

We sincerely appreciate the effort of the AC and all reviewers. To assist the AC and other future readers, we provide summary of our reviews below :

---
**Reviewers' Consensus**

The reviewers mainly appreciated the **novelty, strong performance, and theoretical findings** of our work. We highlight some key strengths pointed out by our reviewers:
* **Novel observation and findings:** Acknowledged by Reviewers FLTy, znnZ, and cQ2e.
* **Theoretical novelty and soundness:** Acknowledged by Reviewers znnZ and cQ2e.
* **Significant speedup performance across various domains:** Acknowledged by Reviewers znnZ and cQ2e.
* **Drop-in practicality :** Acknowledged by Reviewers FLTy and znnZ.

---
**Reviewers' Concerns**

Nevertheless, the main concerns raised were as follows :

**Major Concerns**
* **(FLTy, znnZ) Coupler Overhead:** The major concern of Reviewers FLTy and znnZ regarded the **computational and memory overhead** of implementing coupled sampling, also requesting an analysis of overhead.
  - ***How we addressed :*** We clarified that our coupler has **almost no overhead (under ~4%)** via detailed breakdown experiments. Moreover, we clarified that in an actual efficient implementation, there is virtually no overhead in terms of either time or memory complexity, because our coupler's computational operations are identical to those of standard SJD. We revised our manuscript accordingly (Tab. 4, Sec. 5.2).

* **(cQ2e) More Empirical Data & Theory Discrepancy:** The major concern of Reviewer cQ2e related to the narrative of the paper. Specifically, reviewer noted that (1) Fig. 2 appeared insufficient to support Observation 1 (that low draft token distance tends to yield low total NFEs), and (2) questioned the  discrepancy regarding why the "theoretically sub-optimal" $\pi_{GS}$ sometimes outperforms the "theoretically optimal" $\pi_{MC}$, requesting deeper analysis.
  - ***How we addressed (1):*** We strengthened the empirical support for Observation 1 by adding new experiments in Sec. 5.1 and Fig. 6. By plotting the mean token difference against total NFEs for SJD, MC-SJD, and their mixture in same plot, we demonstrated a strong correlation, confirming that minimizing the expected token difference directly reduces total NFEs.
  - ***How we addressed (2):*** We clarified that this stems from a *misunderstanding*; the paper's claim of "optimality" applies only to **local 1-step collision probability**, rather than global optimality regarding total NFE. Since both couplings ($\pi_{GS}$, $\pi_{MC}$) share similar coupling costs, it is natural that one may slightly outperform the other in specific cases. We argue the fact that two distinct coupling implementations with similar collision probabilities showing comparable performance is actually a strength, serving as empirical evidence for our theoretical foundation.
    - Moreover, in Sec. 5.1 and Fig. 8, we discussed about when and why $\pi_{GS}$ can outperforms $\pi_{MC}$. We showed that GS can result in smaller token differences in the multi-step ($N>1$) regime, both theoretically and empirically, and discuss that this is beneficial for some tasks where draft prediction is relatively easy.

---

The other concerns raised by the reviewers include:

**(Minor) Concerns**

**Reviewer FLTy**
* **Limited Baselines:** We clarified that our baselines are **not limited** because SJD and GSD are the **only existing training-free SD methods for Image AR**. Moreover, even compared to training-based SD methods(EAGLE), our method still achieves SOTA performance, as shown in Fig. 1 and our response.
* **Top-K / CFG Losslessness:** We clarified that our method strictly **guarantees losslessness** even when Top-K sampling or CFG is applied, just like all other SD frameworks.

**Reviewer cQ2e**
* **More Experimental Data:** Reviewer cQ2e requested more experimental data reporting both coupler settings ($\pi_{MC}$, $\pi_{GS}$) with expanded datasets. We conducted all requested experiments(Tab. 1,2,3,5,6), revised our manuscript, and show that our method still surpass standard AR and SJD by a large margin in various settings.
* **Limitation Section:** We have added a limitation section to our manuscript.

**Reviewer znnZ**
* **Failure Modes:** Reviewer znnZ asked if any failure modes were observed. We clarified that our method may fail in one-hot-like distributions, such as text, but this is beyond the scope of this paper.
* **CFG Scale Relation & Adaptive CFG:** Reviewer znnZ requested to quantify the trade-off involving the CFG guidance scale and suggested develop an algorithm to leverage this. We reported the requested data and show that our method can be further improved by scheduling the guidance scale from high to low.

---

We believe all issues have been properly addressed, and we have revised the manuscript to include all changes. Thank you for your time and consideration.

---

### Meta-Review · Area_Chair_jXdE · 2026-01-18

**Summary:**

This paper introduces a new training-free speculative Jacobi decoding method for autoregressive image generation, which outperforms existing methods, such as SJD and GSD. This paper receives 6,4,4 scores and main concerns from reviewers are (1) comparisons with other methods (Reviewer FLTy), (2) Batched Inference (Reviewer FLTy), (3) extra computational and memory overhead (Reviewer znnZ and FLTy), (4) Lack of experiments (Reviewer cQ2e), (5) Limited empirical support for motivation (Reviewer cQ2e), (6) apparent discrepancy between theory and results (Reviewer cQ2e). Considering the reviews and author responses, I suggest rejecting this paper.

**Reviewer Concerns:**

Most concerns (e.g., lack of limitations, limited empirical support, extra computational and memory overhead, analysis of the optimality of MC and GS) are well-addressed.
But there are still some critical concerns about the experiments (1) comparisons with other methods (mentioned EAGLE3 and other state-of-the-art training-based methods are missing) and (2) batched inference.

**Reviewer Scores:**

Reviewer cQ2e may increase the socre.

---

### Decision · Program_Chairs · 2026-01-26

Reject